

# Synoptic- to meso-scale circulation connects fluvial and coastal gravel conveyors and directional deposition of coastal landforms in the Dead Sea basin

Haggai Eyal[1,2], Moshe Armon[1,3], Yehouda Enzel[1], Nadav G. Lensky[2,1]

[1]The Freddy & Nadin Herrmann Institute of Earth Sciences, The Hebrew University of Jerusalem, The Edmond J. Safra Campus, Givat Ram, Jerusalem 91904, Israel

[2]Geological Survey of Israel, 32 Yesha'yahu Leibowitz, Jerusalem 9371234, Israel

[3]Institute for Atmospheric and Climate Science, ETH Zurich, 8092 Zürich, Switzerland

*Correspondence to:* Haggai Eyal (haggai.eyal@mail.huji.ac.il) and Nadav G. Lensky (nadavl@gsi.gov.il)

**Abstract.** Streams convey coarse-clastic sediments towards coasts, where interactions with deltaic and coastal processes determine the resultant landscape morphology. Although extracting hydroclimatic signals from landscapes is a desired goal, many studies rely on interpreting paleoclimatic proxies and the link between depositional/geomorphic processes and the hydroclimate remains vague. This is a consequence of the challenge to link processes that often are studied separately, span across large spatial and temporal scales including synoptic-scale hydroclimatic forcing, stream flows, water body hydrodynamics, fluvial and coastal sediment transport, and sedimentation. Here, we explore this chain of connected processes in the unique setting of the Dead Sea basin, where present-day hydroclimatology is tied closely with geomorphic evolution and sediment transport of streams and coasts that rapidly respond to lake-level fall. We use a five-years-long (2018-2022) rich dataset of (i) high-resolution synoptic-scale circulation patterns, (ii) continuous wind-wave and rain-floods records, and (iii) storm-scale fluvial and coastal sediment transport of varied-mass, 'smart' and marked boulders. We show that Mediterranean cyclones approaching the eastern Mediterranean are the main circulation pattern that can provide sufficient rainfall and winds that concurrently activate two perpendicular sediment conveyors: fluvial (floods) and coastal (wind-waves). The synoptic-scale westerlies (>10 m s$^{-1}$) are orographically funneled inside the Dead Sea rift valley, turning into surface southerlies. They generate 10-30 high-amplitude northward propagating storm waves per winter, with <4 m wave height. Such storms transport cobbles for hundreds of meters alongshore, north of the supplying channel mouths. Towards the decay of the storm wave, the high-altitude synoptic westerlies provide moisture to generate 4-9 flash-floods, delivering unsorted coarse gravels into the basin. These gravels are dispersed alongshore by waves only during subsequent storms. As storm waves dominates and are >five times more frequent than flash-floods, coarse-clastic beach berms and fan-deltas are deposited preferentially north of channel mouths. This depositional architecture, controlled by regional hydroclimate, is identified for both the modern and Late Pleistocene coast and delta environments, implying that the dominance of present-day Mediterranean cyclones has persisted in the region since the Late Pleistocene when Lake Lisan occupied the basin.





## 1. Introduction

Streams and coasts interact and convey coarse sediments. Streams deliver coarse-clastic sediments towards the coast, where the interactions with coastal processes and sediment redistribution in the basin determine deltaic and coastal geomorphology and sedimentology (Ashton et al., 2013; Galloway, 1975; Postma, 1995). While modern deltas and coasts are desired areas for settlements, agriculture, and industry (e.g., Syvitski et al., 2009), ancient deltaic and coastal successions are potential reservoirs of hydrocarbons and water (e.g., Elliot, 1986). Globally, such reservoirs are formed also under receding water levels when the continental shelf and/or slope are exposed, triggering evolution of streams in response to base level fall, and coarse sediment delivery from highstand to lowstand deltas (e.g., Blum et al., 2013) (Fig. 1). Despite the importance of understanding common controls over these jointly operating coarse-clastic conveyors, they are commonly studied separately.

Deltaic architecture is defined on the one hand, by the fluvial regime depending on the hinterland characteristics of the watershed, where climate generates flows carrying sediment load into basins. On the other hand, sediment redistribution and deposition are dictated by the shape, size, bathymetry of a basin, and by the hydrodynamics of waves, currents, tides, and the rate of level changes of the water body occupying the basin (see Fig. 1 in Coleman and Prior, 1982; Postma, 1990; Elliot, 1986). This wide range of influencing factors results in diverse types of deltaic depositional configurations (Postma, 1990, 1995), from which it is challenging to decode hydroclimatic and environmental signals, even in modern environments and more so from past sedimentary records (Hansford and Plink-Björklund, 2020).

In modern *fluvial sediment conveyors*, atmospheric circulation patterns (CPs) and their association with rainfall and floods are extensively studied for specific watersheds and regions (e.g., Bárdossy and Filiz, 2005; Steirou et al., 2017; Merz et al., 2021; Kahana et al., 2002). However, linking the CPs with sediment transport is lacking. A separate body of research deals with flows in channels, their resultant bedload sediment transport (e.g., Reid et al., 1985; Wang et al., 2015; Lekach and Enzel, 2021), channel morphology (e.g., Montgomery and Buffington, 1997), and channel mouth deposition (e.g., Bridge, 1993; Wright, 1977; Coleman and Prior, 1982). In modern *coastal conveyors*, along the shores of oceans or lakes, only a small number of studies have associated CPs with wave climates (Pringle et al., 2014, 2015; Solari and Alonso, 2017; Graf et al., 2013), few of them also attributed these processes to either longshore transport of sand (e.g., Goodwin et al., 2016), or shoreline erosion (Meadows et al., 1997; Pringle and Stretch, 2021). This small body of research stems from the complex link between synoptic-scale circulation, waves, and their resultant sediment transport; processes occurring over a wide range of spatiotemporal scales (Pringle et al., 2015, 2014, 2021; Solari and Alonso, 2017). Therefore, our knowledge regarding the joint fluvial and coastal environments is fragmented, i.e., full linking of the chain of processes/environments, from the synoptic-scale circulation conditions that generate rainstorms-floods, to wind-waves and to sediment transport and deposition in each of the sediment conveyors and their interactions, is missing.

The modern Dead Sea (see regional setting in the next Sect.) is a unique environment providing a "natural laboratory" to study these processes. It has several advantages: (i) The small to medium-scale watersheds ($10^1$-$10^3$ kms) surrounding the lake (e.g., Enzel et al., 2008; Zoccatelli et al., 2019) enable to deeply study the relative impact of different CPs on water discharge (Enzel et al., 2003; Kahana et al., 2002; Dayan and Morin, 2006) and sediment delivery to the basin (Armon et al., 2018; Ben Dor et al., 2018; Armon et al., 2019). (ii) Fluvial and coastal geomorphic responses occur rapidly in response to lake-level fall, enabling a study of real-time





geomorphic processes and present-day sedimentary accumulation under forced regression and known
environmental forcing with implications to the sedimentary record (e.g., Bartov et al., 2006; Sirota et al., 2021).
(iii) Its sedimentary fill is accumulated and well-preserved in a terminal basin, thus it is extensively used to
reconstruct recent limnology and regional paleoclimatology-paleohydrology (e.g., Torfstein et al., 2015, 2013;
Huntington, 1911; Neugebauer et al., 2016; Kiro et al., 2017; Palchan et al., 2017; Ahlborn et al., 2018; Ben Dor
et al., 2018). Despite these advantages, interpretations are still mainly inffered based on selected specific proxies
and the geomorphic processes that led to deposition and their actual link to hydroclimate remains vague.
Armon et al., (2018) have linked the rain- and flood-generating CPs and the resulted sediment plumes dispersed
over the Dead Sea. Linking such sediment dispersion under the lake hydrodynamics is still missing, especially of
coarser sediments. Focusing on gravelly sediments, Eyal et al., (2019) established the recent evolution of an
incising stream transporting increasing volumes of gravelly sediment across the Dead Sea shelf. Then, from the
channel mouth, these coarse sediments are transported and sorted alongshore at the nearshore environment under
seasonal storm wave climates, forming well-sorted coastal landforms (Eyal et al., 2021). However, the
spatiotemporal interactions between the stream and coast and the linkage to or the control of the regional and
synoptic scale hydroclimatology needs elaboration to determine the chain of processes.
Therefore, we study here present-day climatic controls on coarse fluvial and coastal sediment transport by means
of rain, floods, wind, and waves data from the Dead Sea region. We explore the interactions between streams, the
coast and the actively forming coarse-clastic sedimentary record (Fig. 1). We search for the specific hydroclimatic
events controlling the formation of modern geomorphic/sedimentological record and potential insights when
interpreting similar past deposits. We use a five-years-long (2018-2022) dataset comprised of (i) high-resolution
synoptic-scale circulation conditions, (ii) continuous, wind-wave, and rain-floods records, and (iii) storm-scale
fluvial and coastal sediment transport measurements of 'smart' and marked boulders varying in mass. The
manuscript deals with the following questions:
(1)  What is the nature of atmospheric CPs and hydrometeorological conditions activating these fluvial and
coastal conveyors?
(2)  What are the hydroclimatic thresholds in terms of intensity-duration of the rain, and the magnitude of the
floods, winds and waves for transport and deposition of coarse gravel in this currently regressive lake?
(3)  How do rain-producing floods and wind driven waves interact to generate a coastal geomorphic record
with a specific sedimentary architecture?
(4)  What can we learn from the modern sedimentary environment formed by the two conveyors on past
geomorphic records?



Earth **Surface**
**Dynamics**
Discussions

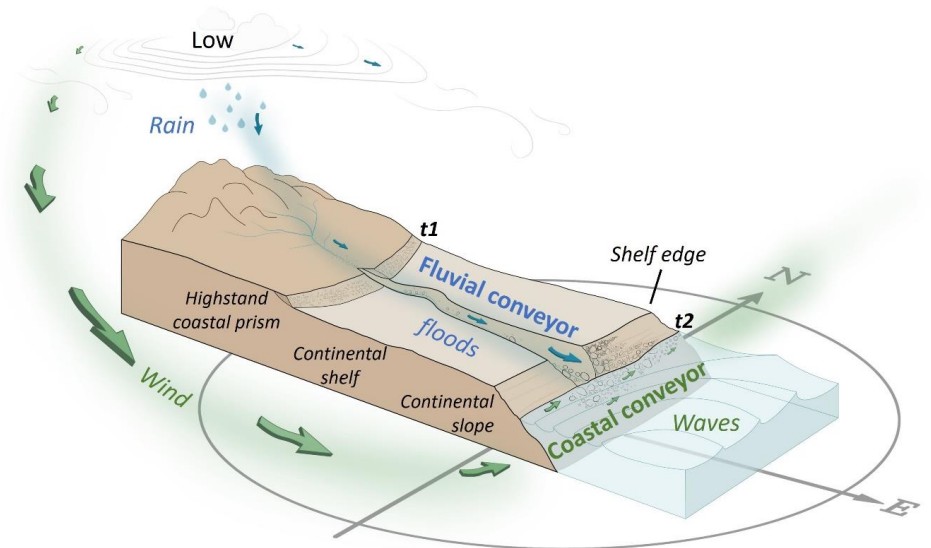

**Figure 1: Schematic illustration of the concepts of sediment transport via the stream and coast explored in this study. The**
**forcing/initiation is at the largest scale; low-pressure atmospheric circulation pattern activates both the fluvial sediment**
**conveyor by generating rainstorms and floods that transport coarse sediments into a receding basin (blue), and the coastal**
**sediment conveyor, in which wind-driven waves obliquely attack the beach and generate longshore sediment drift (green).**
**We discuss the dynamic case during water level lowering. t1 and t2 denote the position of highstand and lowstand**
**shorelines. In the case of the Dead Sea t1 represents the middle of the 20th century and t2 the 21st century.**
## 2. The Dead Sea Regional settings
The Dead Sea basin is a narrow depression, 150 km long and 15–20 km wide, extending south-north (Fig. 2a)
along an actively subsiding tectonic basin of the Dead Sea transform (Garfunkel and Ben-Avraham, 1996). Since
the late Miocene, the basin is occupied by lacustrine water bodies, expanding and contracting due to climatically-
induced water balance and the physiography of the basin (e.g., Zak, 1967; Neev and Emery, 1967; Bartov et al.,
2002; Manspeizer, 1985). During wet and dry climates, the lake level rose and fell, and its area extended and
contracted, respectively (e.g., Bartov et al., 2003, 2006; Bookman et al., 2006; Enzel et al., 2003). The fluvial and
coastal geomorphic responses to these fluctuating lake levels have left well-preserved fan-deltas, paleo-shorelines,
and mudflats, related to the Late Pleistocene Lake Lisan (Bowman, 1971; Amit and Gerson, 1986; FROSTICK
and REID, 1989; Abu Ghazleh and Kempe, 2009) and the Holocene Dead Sea (Enzel et al., 2006., and chapters
in Enzel and Bar-Yosef, 2017) (Fig. 2a).
### 2.1 Geomorphic evolution of streams and coasts in response to shelf and slope exposure
The anthropogenically-induced level decline of the modern Dead Sea, at >1 m y$^{-1}$ (Lensky et al., 2005), due to
water diversions, results in exposure of landscapes considered as fast-forming analogs to the eustatic emergence
of continental shelves and slopes (Dente et al., 2017, 2018; Eyal et al., 2019). The Dead Sea shelf and slope are
mainly comprised of clay silt, laminated, lacustrine deposits over which streams (e.g., Dente et al., 2017, 2018,



2021; Ben-Moshe et al., 2008; Bowman et al., 2010; Eyal et al., 2019) and coasts (e.g., Bowman et al., 2000;
Bookman et al., 2006; Eyal et al., 2021; Enzel et al., 2022) rapidly evolve and can be studied at the field scale in
real-time and at storm- to multi-year resolutions. In the north-western edge of the lake, at the lower reach of the
well-studied ephemeral stream of Nahal (wadi) Og (Fig. 2b-d), hydrological connection with the fast-receding
coastline is maintained by a cross-shelf incision and elongation. Channel bed steepens (channel slope >1.1%),
narrows, and thus increased volumes and clast sizes of coarse sediment are transported to the receding shoreline
intensify with time (Eyal et al., 2019). Gravels are comprised of carbonates and some chert and their intermediate
axes length range between 0.05-0.4 m. From the tributary mouth, the unsorted fluvially-derived sediments are
then transported northward and sorted along the shore, under winter storm waves (Figs. 1, 2d). This process was
measured, quantified, and modelled at the individual storm scale, determining that the coastal longshore sorting
is a direct manifestation of wave climate (Eyal et al., 2021). The interplay between fluvial sediment supply and
longshore transport during winters and significant lake-level decline during summers, results in an annual
separation between individual beach berms which practically, are fossilized, thus preserving their original coastal
sorting (Eyal et al., 2021); i.e., there is no reactivation by subsequent storm waves of the coastal sediments, as
occurs in most shores of earth.

## 2.2 Hydroclimate

### 2.2.1 The potential synoptic-scale climatic drivers at the eastern Mediterranean

Four major seasonal synoptic systems prevail in the eastern Mediterranean during wind and rain storms that affect
the Dead Sea region:
(i)  In winter (mainly December-February), Mediterranean cyclones (MCs) (e.g., Alpert et al., 1990a), also
termed Syrian or Cyprus lows, depending on the respective location of their centers, dominate the stormy
weather (Alpert et al., 1990a; Alpert and Shay-El, 1994). These extratropical cyclones draw moisture
from the Mediterranean and convert it into moderate rainfall over broad areas (e.g., Ziv et al., 2015;
Kushnir et al., 2017). At the regional scale, during the passage of these storms, winds are generally
changing from easterlies into westerlies.
(ii)  In autumn (October-December), Red Sea troughs (RSTs) are most common (e.g., Kahana et al., 2002),
while their "active" variant generates localized and intense rainfall with high spatial variability (Armon
et al., 2018, 2019, 2020; Dayan and Morin, 2006; Belachsen et al., 2017; de Vries et al., 2013; Tsvieli
and Zangvil, 2007). The non-active RST usually brings dry easterly winds at the surface (Saaroni et al.,
1998).

(iii)  In spring (March-May), Sharav lows are frequent in the southeastern Mediterranean (Northern Egypt and
Israel), generating warm and dusty winds (e.g., Alpert and Ziv, 1989) with rarely occurring rains and
high velocity westerly winds following their passage over the area.
(iv)  In summer (June-September), the Persian trough (PT) prevails; low pressure trough extending from the
Persian Gulf to the northeast, along with a subtropical high that borders it from the southwest (Alpert et
al., 1990b); rainfall is scarce as large-scale atmospheric subsidence dominates the region (Rodwell and
Hoskins, 1996; Goldreich, 2003; Kushnir et al., 2017; Tyrlis and Lelieveld, 2013; Lensky and Dayan,
2015), and winds are rather consistently flowing from the north-west (e.g., Tyrlis and Lelieveld, 2013;
Dayan et al., 2017).





### 2.2.2 The fluvial sediment conveyor

Most of the precipitation that produces flash-floods in the Dead Sea region occurs in the heart of the winter, while the full wet season lasts from October to May (Fig. 3a). Annually, the region experiences approximately 20 MCs during winter and early spring with rainstorms typically lasting 2–3 days (Alpert et al., 2004a; Saaroni et al., 2010) generating relatively high-volume floods (Enzel et al., 2003, 2008; Kushnir et al., 2017; Armon et al., 2018; Shentsis et al., 2012). Smaller number of rainstorms during the autumn and spring are usually associated with ARSTs (Armon et al., 2018).

The western water divide of the larger Dead Sea tributaries is at the Judean Mountains with peaks up to ~1000 meters above sea level (masl) and Mediterranean/semi-arid climate (Fig. 2b). From the water divide eastwards, the topography steeply slopes down to the Dead Sea at elevation of ~437 meters (in 2022) below sea level (mbsl) over a short distance of ~30 km, resulting in a sharp climatic gradient (Fig. 3a) due to the orographic rain-shadow effect (Goldreich, 2003; Kushnir et al., 2017). Thus, streams draining into the Dead Sea from the west are ephemeral and are subjected to flash-floods during sufficient storm rainfall (e.g., Morin et al., 2009). For example, in the Nahal Og watershed (137 km$^2$), the climatic gradient ranges from >500 mm y$^{-1}$ in the western headwaters to as low as ~50 mm y$^{-1}$ at the Dead Sea shore (Figs. 2b, 3a). The mean annual total rain volume falling over the basin is ~40x10$^6$ m$^3$y$^{-1}$ (Haviv, 2007; Ben Moshe et al., 2008), of which only a small portion reaches the lake. The highest peak discharge estimated for the stream by high-water marks after the rare flood of 2006, is 330 m$^3$ s$^{-1}$ (Arbel et al., 2009). In Eyal et al., (2019), direct observations of flow marks at a specific location along the channel were interpreted to represent the peak discharge of the common floods of ~20 m$^3$ s$^{-1}$. Floods, lasting from a few hours and up to a day, are generally short and quick response to high-intensity rain (e.g., Morin et al., 2009).

### 2.2.3 The coastal sediment conveyor

Winds along the Dead Sea have a bimodal directional distribution of either northerly or southerly direction (Fig. 3b,c) affected by the steep orography and north-south elongation of the Dead Sea rift (Bitan, 1974, 1976; Segal et al., 1983; Vüllers et al., 2018; Kunin et al., 2019). During summer, the diurnal cycle dominates with dry and warm northerly winds (<10 m s$^{-1}$) blowing stronger at night-time and weaker during the day, attributed to the meso-scale circulation of the Mediterranean Sea breeze (Alpert et al., 1997; Gertman and Hecht, 2002; Lensky and Dayan, 2012; Lensky et al., 2018; Hamdani et al., 2018; Kunin et al., 2019; Naor et al., 2017). During winter, the diurnal cycle is less dominant as the above-mentioned synoptic scale circulation governs (Hamdani et al., 2018) with southern windstorms, <20 m s$^{-1}$, lasting from a few hours to three days, blowing over the ~40 km south-to-north lake fetch (Eyal et al., 2021). These high-magnitude winter windstorms generate waves with a maximum height of ~4 m, wave periods of ~4 s, and wavelengths of ~25 m in the northeastern shores of the Dead Sea (Eyal et al., 2021). During storms, waves approach the coast at ~45° (Eyal et al., 2021), forming optimum conditions for unidirectional longshore drift (Longuet-Higgins, 1970; Van Hijum and Pilarczyk, 1982; Ashton and Giosan, 2011). Along the waterline of the Nahal Og coast, fluvially-derived gravels are distributed over a 20–30 m wide strip, covering the lake floor by a monolayer, extending to a water depth of ~2.5 m; at this depth, a transition to sandy-silty wave ripples is documented. The longshore transport and sorting of the coarse gravel and their link to the wave climate were presented in Eyal et al., (2021) for three intensively-monitored storms.



Earth **Surface**
Dynamics
Discussions

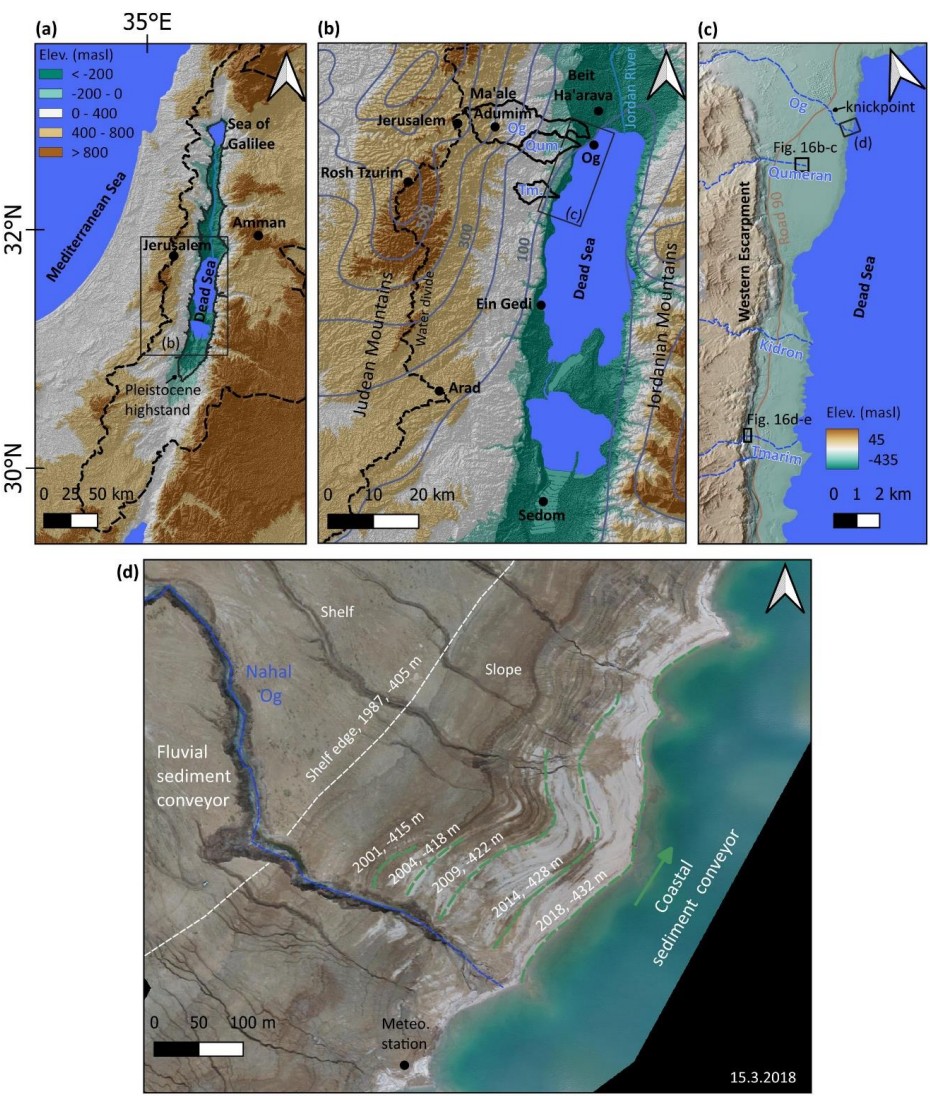

Figure 2: Regional setting. (a) The eastern Mediterranean; shown are the Dead Sea watershed (black dashed line) and the highstand of the Late Pleistocene Lake Lisan, the predecessor of the Dead Sea (black line). (b) The Dead Sea region. Shown are the regional water divide of the Judean Mountains (dashed black line) and the watersheds of the studied tributaries: Og (Og), Qumeran (Qum.) and Tmarim (Tm.) (black polygons). Grey contours are isohyets (mean annual precipitation in mm y⁻¹). They present the rain shadow of the Judean Mountains towards the Dead Sea valley. Black dots are meteorological stations used in this study. (c) The tributaries draining into the north-western Dead Sea (blue dashed lines) and the Dead Sea western escarpment. (d) Aerial photograph of the lower reach of Nahal Og emphasizing the fluvial and coastal conveyors; note the increasing extension farther north, from the stream mouth, of the coastal gravel with lowering of the lake (green lines). It should be stressed that the tributaries north of Nahal Og drain the mudflat and do not carry gravel. Modified from Eyal et al., 2021.









**Figure 3: Rainfall and wind forcing during the five, intensively measured hydrological years: December 2017- June 2022. (a) Daily (bars, left-axis) and seasonal cumulative (lines; right-axis) rainfall measured, from west to east, in Jerusalem (blue), Ma'ale Adumim (orange), and Beit-HaArava (yellow), representing the headwaters, the center, and lower areas of the watershed, respectively (stations locations are presented in Fig. 2b). Vertical black lines are occurrences of floods (Table S1 in the supplement). Note that most storms affect the entire region with consistent decline in rainfall amounts away from the water divide. (b) 10-minutes (blue crosses) and daily average (orange line) wind speed at Nahal Og mouth. Windrose for (c) Nahal Og (-430 masl) and (d) Jerusalem (835 masl) representing the frequency and directionality of winds during the study period. Note the orthogonal wind directions; in the upper watershed it is dominated by westerlies, while at the same time, within the Dead Sea rift valley, it is dominated by northerlies and southerlies.**

## 3. Methods, data, and analysis

To unfold the chain of processes from the synoptic-scale climatology to rainstorms and flood hydrology and to wind and wave climate, which are involved in the formation of the coastal sedimentary record along this regressive lake, we assembled a high-resolution, rich dataset. It is comprised of: (1) Five-year long, continuous monitoring of winds, waves, lake level, rain and flood hydrology. (2) Storm-scale sediment transport documented in the channel and shore. (3) A combination of this dataset with atmospheric CPs using atmospheric reanalysis. These observations constitute a one-of-a-kind dataset of coeval processes at such a resolution, undoubtedly for this region and probably for elsewhere. Additionally, although these observations are based on only five years of data, comparing the rainfall and wind timeseries with adjacent long record weather stations, indicates that these years well represent the mean climatic conditions (Sect. S2 in the supplement).

### 3.1 Field measurements

*Wind* speed and direction at 10-min intervals were (a) measured at the Nahal Og mouth by a Gill-WindSonic sensor located ~5 m above the lake surface, between December 2017 and June 2022, and (b) obtained from the Israel Meteorological Service for the stations of Jerusalem Center (1999-2022), Ma'ale Adumim (2007-2022), Ein Gedi (2007-2021), Rosh Tzurim (2001-2021), Arad (1999-2021), Sedom (1999-2021) and Beit Ha'arava (2008-2022) (Fig. 2b).

*Waves* were measured at 4 Hz frequency by a water pressure sensor (Keller-PAA 36 Xi W) at water depth range of 12 (December 2017) to 8 m (June 2022). Significant wave height and period were analyzed, accounting for the attenuation of wave-induced pressure variation with water depth, and the temporal change of water depth due to lake-level decline (Karimpour and Chen, 2017). From the continuous 4 Hz data, differences between maximum and minimum pressure at 10-min resolution were normalized between 0 (no waves) and 1 (highest observed wave height, H = 4 m) and used as proxies for the significant wave height (Fig. S3, Eyal et al., 2021). This was done as the long time-series of 4 Hz measurements is incomplete. This analysis was validated by 16 Hz measurements of RBR-solo-wave pressure sensor, deployed at 5-m water depth during three storm waves.

*Rain* data at 10-min intervals were obtained from the Israel Meteorological Service for the stations of Jerusalem Center (1999-2022), Ma'ale Adumim (2008-2022) and Beit Ha'arava (2008-2022).

A *Flood Hydrology* data set was gathered from several sources (see Sect. S1 in the supplement), as no direct discharge measurements exist in the watershed: (a) Observations obtained by Time-Lapse Cameras (TLCs) and



real-time field surveys, from which hydrographs were estimated using the manning formula (as in Eyal et al.,
2019) (when high flows occurred at night, high water marks were estimated from the daylight video). (b) Flood
reports obtained from the Israel Flash-flood Forecasting Center, Water Authority of Israel. (c) Flood reports
obtained from the Desert Floods Research Center categorized into no flood, weak flood, moderate flood, and large
flood. (d) Social network reports (e.g., Borga et al., 2019), providing an almost complete binary series of yes/no
flood occurrences and their estimated magnitude. These observations were synthesized to classify the floods into
four categories according to the estimated flood peak-discharge: low-flow floods, which due to transmission losses
do not reach the lake, weak floods, moderate floods, and large floods. Estimation of the extremity of the peak
discharge for each class was evaluated according to Rinat et al., 2021 (their Fig. 8). Cross-checking between the
information sources and close monitoring of the events during the measurement interval of 2017-2022 provides a
high level of certainty about the completeness of the flood time series. However, it must be noted that hydrograph
estimation gives rough values rather than exact high-resolution measurement data.
The *Dead Sea level* was obtained from Water Authority of Israel at a monthly resolution.
*Sediment transport* was measured using boulders with masses ranging between 0.5-100 kg. (a) Tens of boulders
were positioned in the upstream channel before a flood to estimate transport distances by a single event. (b) Along
the beach, using "smart" and painted boulders as described in Eyal et al., 2021, for three different storms.
*Late Pleistocene to modern fan-deltas* were analyzed by: (a) Airborne LiDAR-based DEMs for 2020, with
horizontal and vertical resolutions of 0.5 and 0.25 m pixel$^{-1}$, respectively (obtained from the Geological Survey
of Israel). (b) Orthophoto imagery and georeferenced aerial photographs from the years 1945, 1967, 1980, 1987
(obtained from the Survey of Israel). (c) A satellite image from 1971 (Corona mission, Grosse et al., 2005; data
available from https://earthexplorer.usgs.gov) with a spatial resolution of up to several meters per pixel.  These
images were used to examine landscape change preceding the available LiDAR-based DEMs. They were also
used for mapping and determining the altitude of shorelines of the late 20[th] and 21[st] centuries, recognized on both
air photographs and LiDAR and of Late Pleistocene shorelines in Nahal Tmarim (location in Fig. 2b,c). DEM and
hill shade of 30 m pixel$^{-1}$ resolution obtained from Geological Survey of Israel were used for location maps (Figs.
2a,b, and 10a)
**3.2 Data analysis**
**3.2.1 Storm detection**
Over 120 storm waves were defined according to a physical threshold of the critical wave height for mobilization
of a 1 kg clast: $H_{cr}$=~0.6 m as determined previously by Eyal et al., 2021. A one-day interval was selected as
separating between individual storms. The timing of storm initiation and cessation was obtained using a lower
wave height threshold (e.g., Molina et al., 2019), H=~0.15 m, which is a sufficiently lower value to account for
the entire storm-wave duration (Fig. 4). As the waves are wind-driven (see below Sect. 4), windstorms were
defined according to the timing of the storm waves. This was done by applying the timing of the wave initiation
and cessation to the wind speed timeseries and redefining the windstorm initiation and cessation according to a
wind speed daily mean threshold of 3 m s$^{-1}$ (Fig. 4). This threshold optimally represents the storms following a
comparison with a range of thresholds (0.5 – 5 m s$^{-1}$). The storm peak is defined as the maximal wind value in the
interval between the initiation and cessation. Rainfall was analyzed at hourly intervals, accumulated from the 10
minutes data. Thirty-two flood-producing rainstorms were defined by detecting rainstorm peaks using a one-day



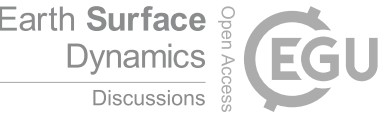

time interval before and after flood initiation. The timing of rainstorm initiation and cessation were redefined
using a 0.1 mm h$^{-1}$ threshold and a separation of at least six hours between successive storms (e.g., Marra et al.,

2020).

**3.2.2 Synoptic classification**

We classified wind-waves-rain storms into four classes representing the most common synoptic circulation
patterns prevailing in the region (Sect. 2.2.1): Mediterranean Cyclones (MCs), Active Red Sea Troughs (ARSTs),
Persian Troughs (PTs), and Sharav Lows (SLs). To do so, we generalized the 19 classes obtained by the semi-
objective synoptic classification introduced by (Alpert et al., 2004b) for the eastern Mediterranean, which is based
on daily (12:00 UTC) meteorological fields at the 1000 hPa pressure level from the NCEP/NCAR reanalysis
(2.5$^0$ spatial resolution). We classified a storm as a MC if one of the storm days was considered as a MC. ARST
was defined if one of the storm days was considered as ARST with no MC prevalence. SL was classified if one
of the days during the storm was classified as SL, regardless of the other classes obtained by the semi-objective
classification. PT was classified only if it appeared in the summer months between June and September (e.g., Ziv
et al., 2004), even if it appeared with other classes. Otherwise, it was classified as a MC in accordance with weak
cyclones manifested as a shallow trough in the north-eastern Mediterranean (Ziv et al., 2022). The 13 cases
classified by the semi-objective classification as highs were manually inspected, and were reinterpreted as MCs,
as they represent the ending of MCs (e.g., Armon et al., 2019; Marra et al., 2021).

**3.2.3 Composite and individual storm CPs**

Composite and individual storm CPs were analyzed using data from the European Center of Medium-range
Weather Forecasts (ECMWF) Reanalysis model 5 (ERA5; Hersbach et al., 2020). Sea level pressure and 10-m
above ground wind maps were produced for the wind-wave storms at their onset, peak and cessation at a resolution
of 0.5° per pixel. Composite maps were obtained for (i) the mean conditions during the different storm parts both
for all CPs together and separately for, (ii) the lowest, intermediate, and highest terciles of the wave energy,
duration, and wave height, and (iii) the climatology of wave-producing CPs, non-wave-producing CPs, and the
anomaly of the wave-producing CP compared to the mean conditions of CP for the same period (2017-2022).





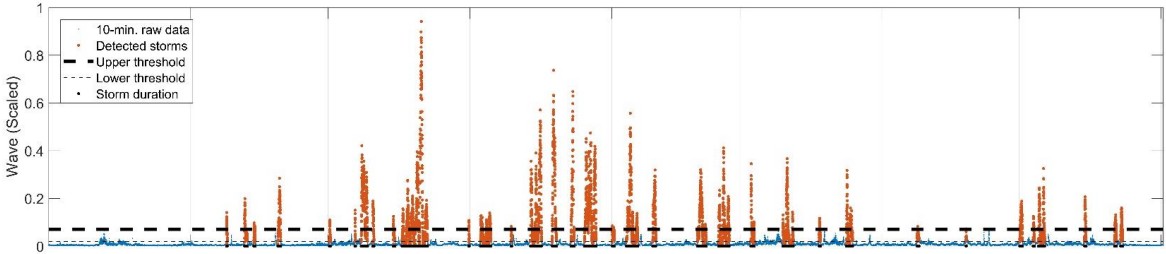

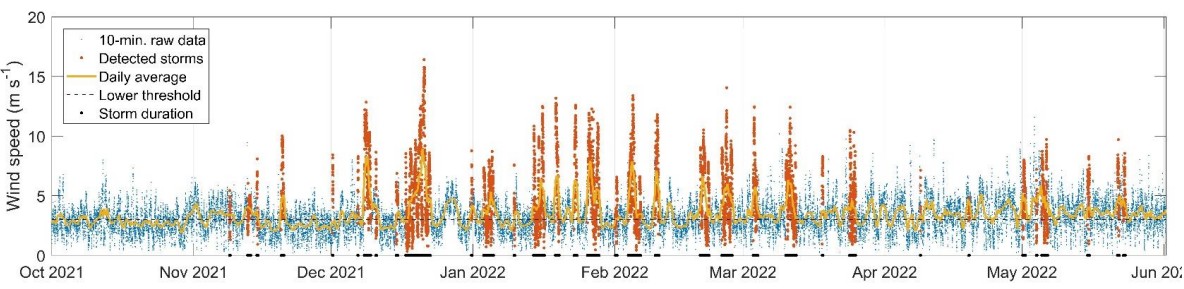


**Figure 4: An example of wind-wave storm detection during one hydrological year (2021-2022). (a) Storm waves (orange**
**dots) were detected by an upper physical threshold following Eyal et al., 2021 (thick dashed black line), with the full**
**duration (black dots marked on the x-axis) defined by a lower threshold (thin dashed black line). (b) Windstorms**
**(orange dots) were defined according to the detected storm waves, with the full duration defined by a lower threshold**
**(dashed black line) following the daily average of the wind speed (yellow line).**



## 4. The fluvial and coastal sediment conveyors and their synoptic-scale hydroclimatic control


We present insights from five representative storm-scale case studies in Sect. 4.1 for which we have detailed
measurements of sediment transport in the stream and coast under the forcing of atmospheric CPs, winds and
waves, rain, and floods (Figs. 5-9). Each component is described with respect to the timeline of a wind-wave
storm from its onset, rise, peak, decay, and cessation. Then, in Sect. 4.2, we present the separation of the wind
field into two levels with perpendicular directions, i.e., the regional surface wind during storms both outside and
inside the Dead Sea rift valley (Fig. 10). In Sect. 4.3 we generalize the processes leading to the activation of the
two sediment conveyors with a full analysis of the wind-wave storms and floods of the past five years with their
synoptic- and meso-scale climatology (Figs. 11-13). Given that MCs stand out as the main activators of the
sediment conveyors (Sect. 4.3 and Fig. 11), we describe the results according to the evolution of this synoptic-
scale CP and add information on other CPs when necessary.

### 4.1 The stream and coast at the storm scale


### 4.1.1 Storm-scale atmospheric CPs


At the onset of the wind-wave storms, the centers of the MCs are located north of the study region: (i) In the
vicinity of Greece, as far as ~1500 km northwest of the Dead Sea (Fig. 5c). (ii) In the eastern Mediterranean near
Cyprus, ~500 km northwest of the Dead Sea (Figs. 6-7c). (iii) In Syria or Iraq, 500-700 km north-northeast of the
Dead Sea (Fig. 8c). Only seldom storms occur when the cyclone is near the Dead Sea, in southern Israel (Fig. 9,
see a more detailed description of the event in Dayan et al., (2021) and in Rinat et al., (2021). The prevailing storm
circulation is of anti-clockwise westerly/south-westerly winds. Towards the storm peak, MCs focus (i.e., become
smaller), deepen, and move eastwards (Figs. 5-8d). In mature and ending stages of MCs, the regional westerly
flow and lowered inversion (Armon et al., 2019; Goldreich et al., 2004) are manifested by 'mountain waves'; i.e.,
south-north elongated cloudy crests extending over the Jordanian mountains and plateau (Fig. 6h). The storm is
over when the low-pressure systems become larger, shallower, move further to the east, and a high-pressure
system invades the region (Figs. 5-8e).

### 4.1.2 Local wind and waves


While at the regional scale westerly flows dominates, at the local scale, over the Dead Sea itself, a sharp rise of
pronounced southern winds characterizes the onset of storms under MCs as measured along the Dead Sea shores
(Figs. 5-9b). With the intensification of the winds to >10 m s$^{-1}$ and up to 20 m s$^{-1}$, northward-propagating waves
also intensify (Fig. 5-9b). At the end of the storm, diverse directionality that characterizes the pre- and post-storm
intervals of the wind (Figs. 5-9b) prevails, and the wind and waves quickly calm down.

### 4.1.3 Rain and floods


Rainfall in the drainage basin (Ma'ale Adumim) initiates coevally with the wind-wave storms, normally
intensifying after the storm wave peak (Figs. 5, 7–9a) or even during the peak (Fig. 6a), reaching moderate to high
intensities relative to this dry climate, of > 5 mm h$^{-1}$ for the duration of at least an hour (Figs. 5–9a). Rainfall
intensity may comprise of several maxima, and accordingly, the flash-flood hydrograph presents several peaks
(Figs. 5, 7, 8a). Flood discharge ranges between weak floods (~5 m$^3$s$^{-1}$) (Fig. 5a), to the largest flood documented





between 2017-2022 with an estimated peak discharge of $120\pm30$ m$^3$ s$^{-1}$ (Fig. 8a). These floods typically last <24
h lagging a few hours after the rain peak; this important observation indicates that sediments are delivered to the
stream mouth towards the decay or end of the storm wave.

### 4.1.4 Sediment transport

With the rise of winds and waves and exceedance of the critical wave height (Fig. 4), certain clasts are mobilized
according to their mass as indicated by the recorded accelerations and rotations (Fig. 6f, Eyal et al., 2021). During
the storm peak, the highest accelerations and rotations are recorded (Fig. 6f). By the end of the storm wave, gravels
are sorted along the shore as the displacement decrease with increasing clast mass, according to a power law (Eyal
et al., 2021) (Figs. 5f, 6g, 9f). Larger clasts weighing ten of kilograms are transported to tens of meters, and finer
clasts weighing kilograms are transported hundreds of meters along the shore (Figs. 5f, 6g, 9f). Coevally, or by
the end of the storm waves, a flood reaches the stream outlet into the Dead Sea (Figs. 5–9a) transporting at a
single, relatively low-discharge flood, cobble-boulder sized clasts, >10 kg, along the incised channel across the
one-kilometer-wide shelf (Fig. 5a). The transport rate of boulders per single event along the shore is one to two
orders of magnitudes smaller relative to the transport in the stream. In the common case of floods that are generated
after the storm wave, delta deposition and sediment progradation of up to 20 m offshore is observed at the channel
mouth (Fig. 9g-i). In such a case, the activity of the coastal conveyor precedes the fluvial conveyor, and longshore
transport and sorting of the fluvio-deltaic sediments can only happen during the next storm. A different case is
when floods do not reach the lake and only the coast is activated by the storm, reworking the sediments delivered
by the previous storms in the season (Fig. 6a).

Earth **Surface**
**Dynamics**
Discussions

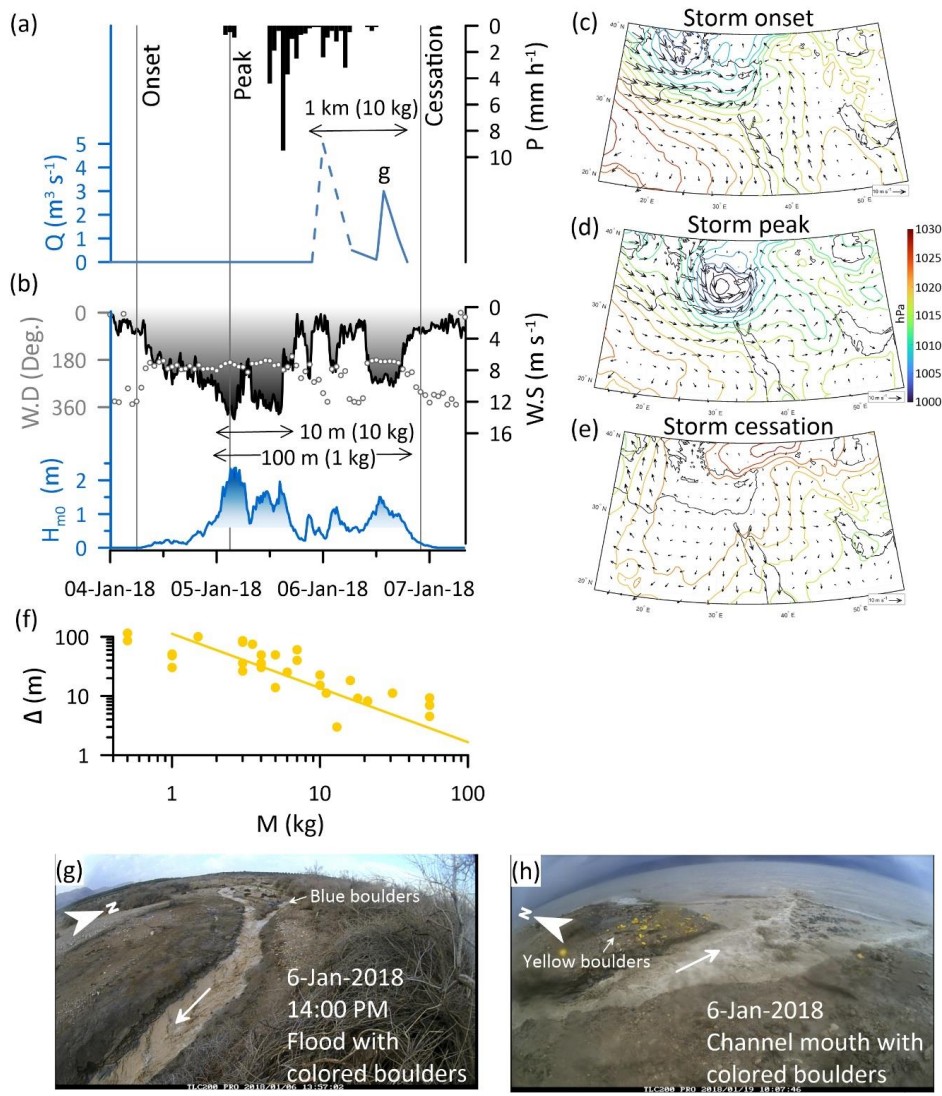

**Figure 5: Storm-scale observations (4-7 January, 2018) of the chain of processes from the synoptic scale atmospheric circulation that generate rainstorms-producing floods, wind-wave storms, resulting in fluvial and coastal sediment transport. (a) Hourly rainfall (P, Ma'ale Adumim, Fig. 2b), flood discharge (Q, solid line based on TLC and dashed line based on high-water marks). During this flood, colored cobbles-boulders were transported across the entire 1 km shelf width into the Dead Sea. (b) Wind (W.S-wind speed, W.D-wind direction in dots) and wave height (H-significant wave height, colored gradient fill indicates waves above transport threshold). (c, d, and e) CP maps of a deep Mediterranean Cyclone plotted according to the onset, peak, and cessation of wind, respectively. (f) Longshore displacement (Δ) of various-mass boulders (M) (yellow dots), transported from the channel mouth northward and sorted alongshore according to a power-law (yellow line), following Eyal et al., 2021. (g) The flood at the stream knickpoint where boulders were colored. (h) The flood flows into the Dead Sea, where coastal boulders are colored.**



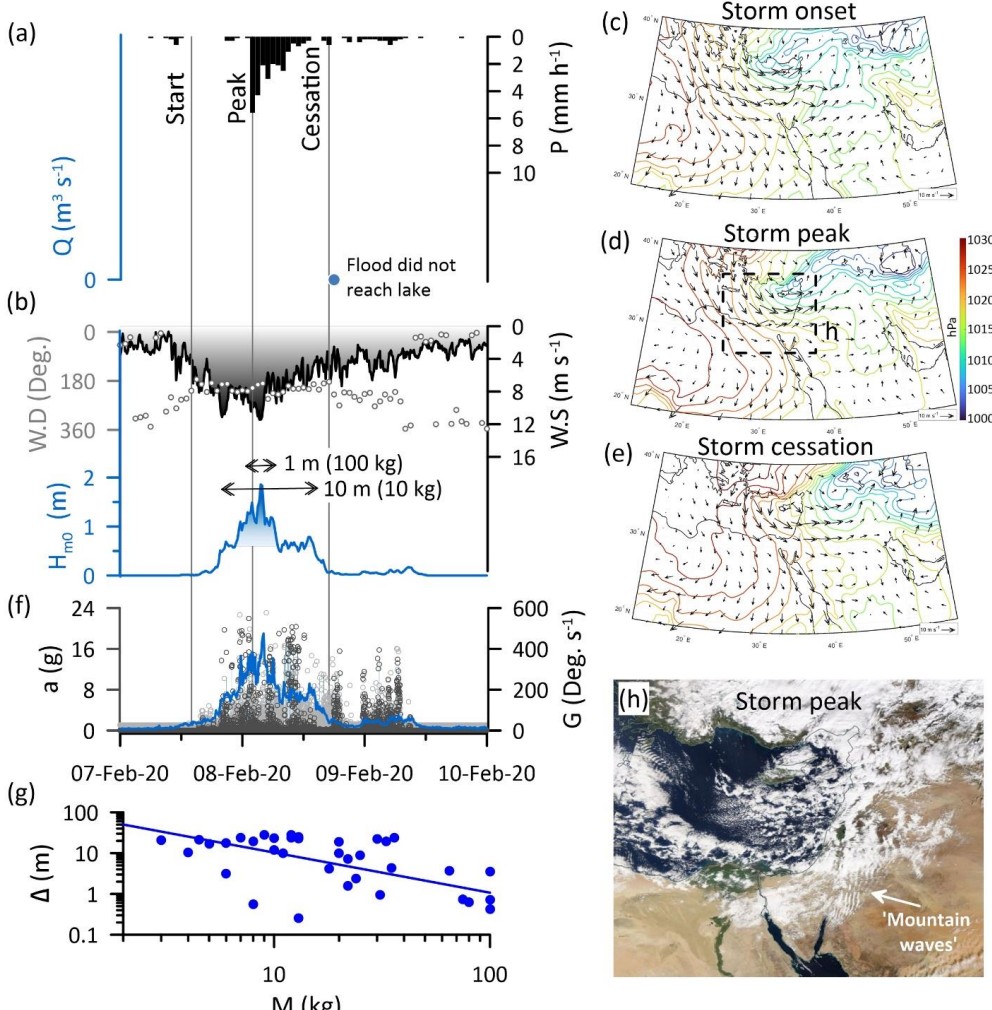

**Figure 6: Storm-scale observations (7-9 February, 2020) of the chain of processes from the synoptic- scale atmospheric circulation that generate rainstorms-producing floods, wind-wave storms, resulting in fluvial and coastal sediment transport. (a) Hourly rainfall (P, Ma'ale Adumim, Fig. 2b), flood was generated but did not reach the lake. The timing of a first wave is marked by a blue dot. (b) Wind (W.S-wind speed, W.D-wind direction in dots), and wave height (H-significant wave height, colored gradient fill indicates waves above transport threshold). (c, d, and e) CP maps of a Mediterranean Cyclone plotted according to the onset, peak, and cessation of wind, respectively. (f) Resultant acceleration (a, grey dots) and rotations (G, black dots) recorded by five, various-mass smart boulders indicating the real-time motions of clasts under storm waves, following Eyal et al., 2021. (g) Longshore displacement (Δ) of various-mass boulders (M) (blue dots), transported from the channel mouth northward and sorted alongshore according to a power-law (blue line). (h) Aerial photograph of the eastern Mediterranean during the storm peak (8 February, 2020) obtained from https://worldview.earthdata.nasa.gov/, location in (d). Note the south-north elongated cloudy crests termed 'mountain waves', indicating on the synoptic westerly air flow.**

Earth **Surface**
**Dynamics**
Discussions

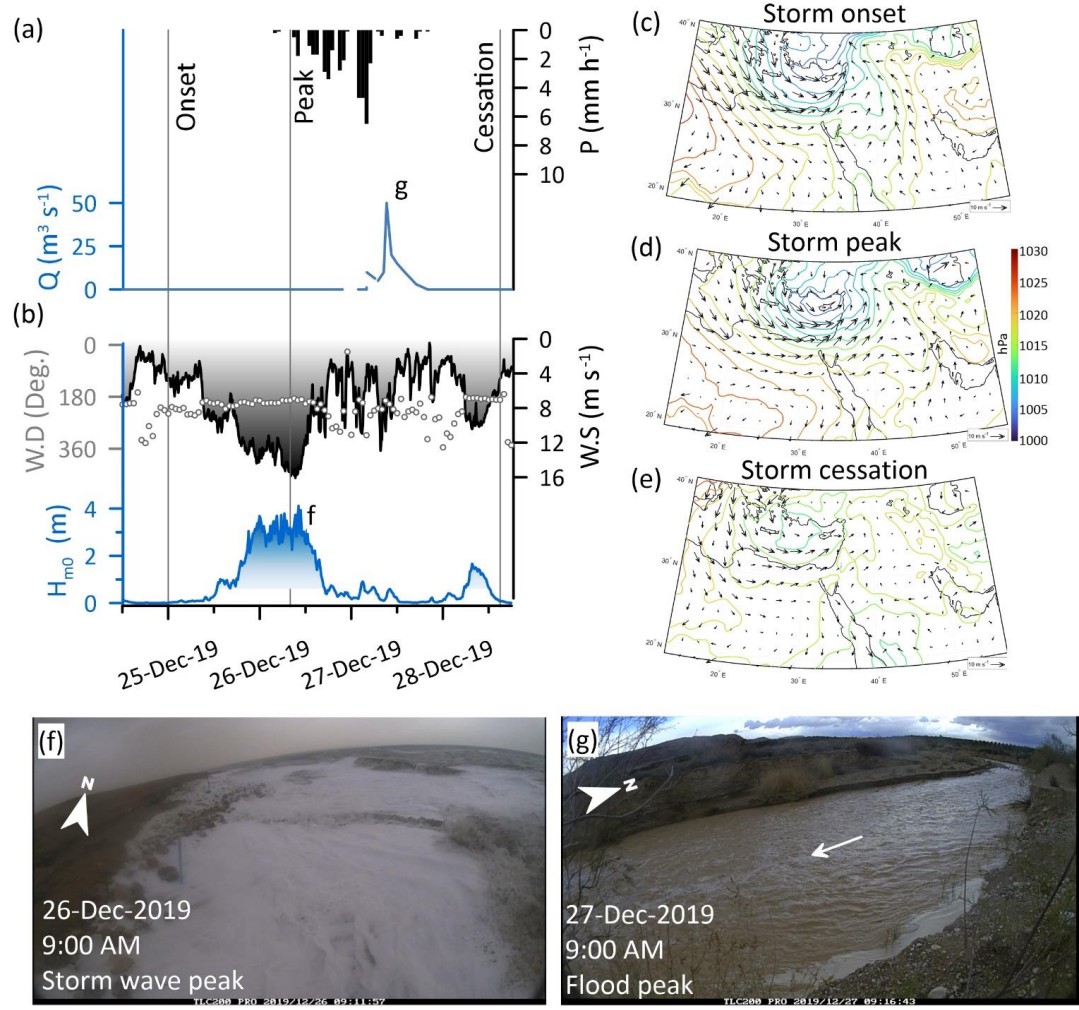

**Figure 7: Storm-scale observations (25-28 December, 2019) of the chain of processes from the synoptic-scale atmospheric circulation that generate rainstorms-producing floods, wind-wave storms, resulting in fluvial and coastal sediment transport. (a) Hourly rainfall (P, Ma'ale Adumim, Fig. 2b), flood discharge (Q, solid line-TLC). Wind (W.S-wind speed, W.D-wind direction in dots) and wave height (H-significant wave height, colored gradient fill indicates waves above transport threshold). This storm wave was the largest documented in our record (Video supplement). (c, d, and e) CP maps of a deep Mediterranean Cyclone plotted according to the onset, peak, and cessation of wind, respectively. (f) The storm wave during its peak, which is the highest in our record. (g) The flood peak downstream to road 90 (location in Fig. 2c).**

Earth **Surface**
Dynamics
Discussions

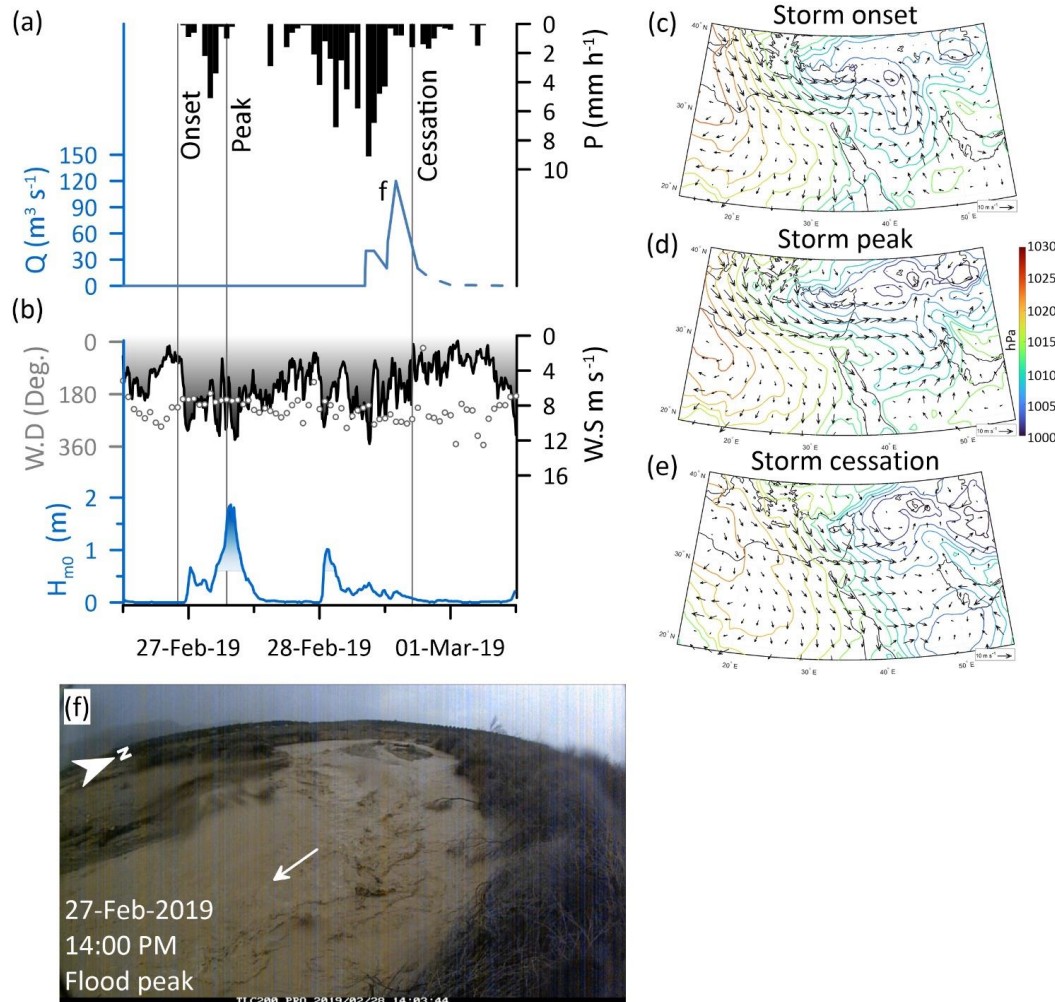

**Figure 8: Storm-scale observations (27-28, February 2019) of the chain of processes from the synoptic scale atmospheric circulation that generate rainstorms-producing floods, wind-wave storms, resulting in fluvial and coastal sediment transport. (a) Hourly rainfall (P, Ma'ale Adumim, Fig. 2b), flood discharge (Q, solid line-TLC). This flood was the largest documented in our record (Video supplement). (b) Wind (W.S-wind speed, W.D-wind direction in dots) and wave height (H-significant wave height, colored gradient fill indicates waves above transport threshold). (c, d, and e) CP maps of a Mediterranean Cyclone centered to the east of the Mediterranean, with an extended trough to the eastern Mediterranean, plotted according to the onset, peak, and cessation of wind, respectively. (f) The flood peak downstream of Highway 90 (location in Fig. 2c).**

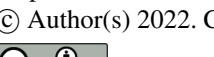

Earth **Surface**
**Dynamics**
Discussions
EGU

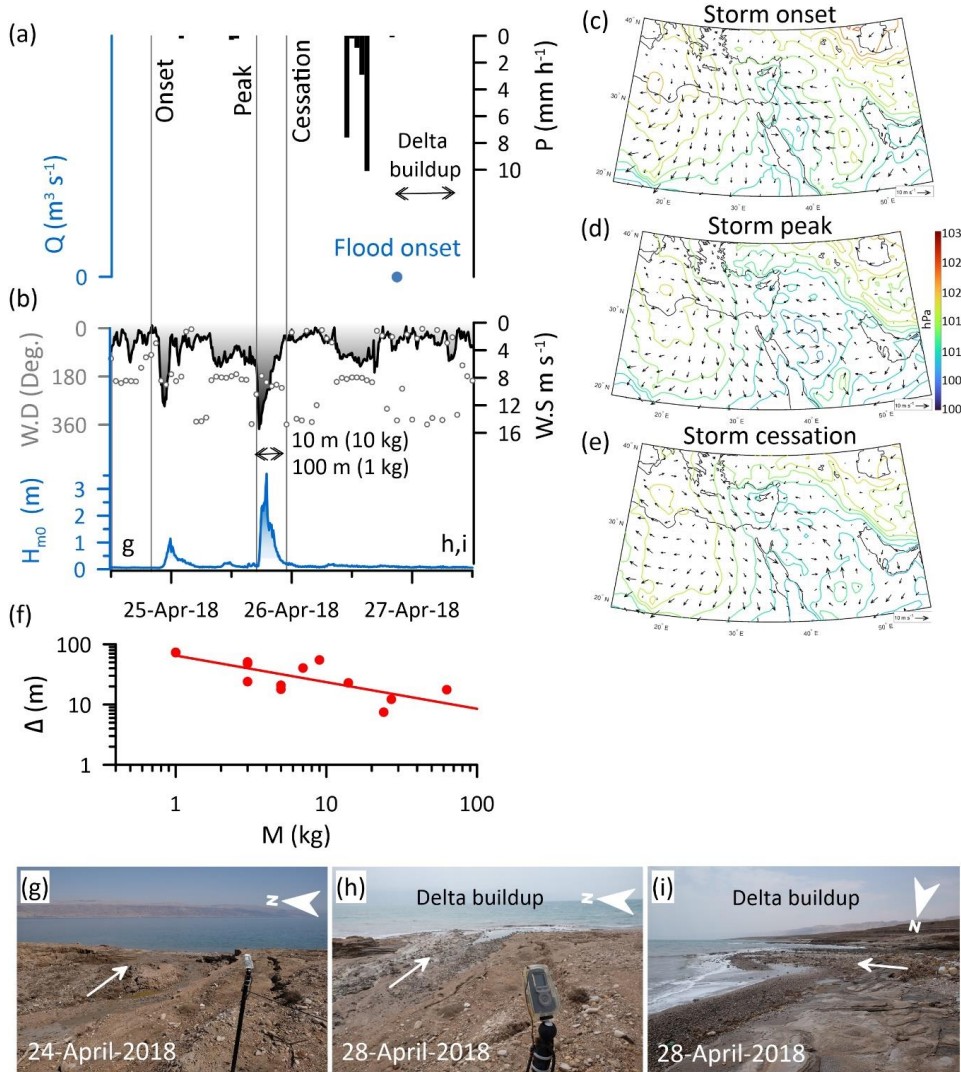

**Figure 9: Storm-scale observations (25-27 April, 2018) of the chain of processes from the synoptic-scale atmospheric circulation that generate rainstorms-producing floods, wind-wave storms, resulting in fluvial and coastal sediment transport. (a) Hourly rainfall (P, Ma'ale Adumim, Fig. 2b). The flood discharge was high, as indicated from a field visit during this storm. (b) Wind (W.S-wind speed, W.D-wind direction in dots) and wave height (H-significant wave height, colored gradient fill indicates waves above transport threshold). (c, d, and e) CP maps of a southern-centered Mediterranean Cyclone plotted according to the onset, peak, and cessation of wind, respectively. This storm also was discussed in detail in Rinat et al., (2021) and Dayan et al., (2021). (f) Longshore displacement (Δ) of various-mass boulders (M) (red dots), transported from the channel mouth northward and sorted alongshore according to a power-law (red line), following Eyal et al., 2021. (g) The channel mouth before the storm. (h and i) The channel mouth after the flood ends with prominent fan-delta progradation of ~20 m offshore.**





**4.2 Synoptic-scale and orographically channelled surface winds activating the two perpendicular sediment conveyors**

During MC storms, synoptic-scale westerly circulation is consistent with measurements of surface wind in ground stations, located along a south-north transect of the 600-1000 masl water divide at the Judean Mountains (Fig. 10a-d). Coevally, a transect of the winds within the Dead Sea rift valley at an elevation of ~400 mbsl, ~30 km east of and sub-parallel to the water divide, indicates that the high-magnitude surface winds have a clear southern directionality (Fig. 10a, e-g). We attribute this directionality change, from the regional westerlies into in-rift valley southerlies during the same individual storm, to the orography-funneling effect of the Dead Sea valley with its south-to-north oriented rift shoulders (e.g., Bitan, 1976). Consequently, we recognize that the winds associated with the main synoptic-scale circulation pattern (MC) splits into two perpendicular directions; these two hydroclimatic generators activate differently the coarse-sediment conveyors (Figs. 1, 10, Video supplement): (i) Westerlies at high altitudes convey moisture from the Mediterranean Sea, with rainfall amounts tending to increase when air parcels encounter the orographic barrier of the Judean Mountains and then decrease in the rain shadow area of the Dead Sea rift valley (Sharon and Kutiel, 1986; Goldreich, 1994; Marra et al., 2022). This orographic effect is an important permanent feature over the last millions of years since the rift reached its shape. This orography determines the amount and distribution of rainfall over the western Dead Sea watersheds and, in turn, the characteristics of floods, and with them the timing of sediment delivery into the basin. The conveyance of moisture continues to the east of the Dead Sea and rainfall amount increases again with the upslope flow over the Jordanian mountains >1000 masl (e.g., Armon et al., 2019); as a result, floods are generated, and sediments are delivered to the Dead Sea from the eastern watersheds at the very end of the storms. (ii) At the surface, southerlies blow perpendicular to and coeval with the synoptic-scale mountainous winds. The meso-scale funneling of winds blowing over the lake results in south-to-north waves propagation and thus, at the coast, the redistribution of sediments preferentially northwards from the channel mouths along the Dead Sea shores.

Weaker CPs have different air trajectories, but as long as the synoptic winds have a slight southern component, the topography and shape of the Dead Sea rift margins govern, resulting in southerly-funneled winds. For example, under ARST conditions, the synoptic scale wind is southeasterly, while the actual surface wind measurements are pure southerlies (Fig. S4).

Earth **Surface**
**Dynamics** Open Access
Discussions

EGU

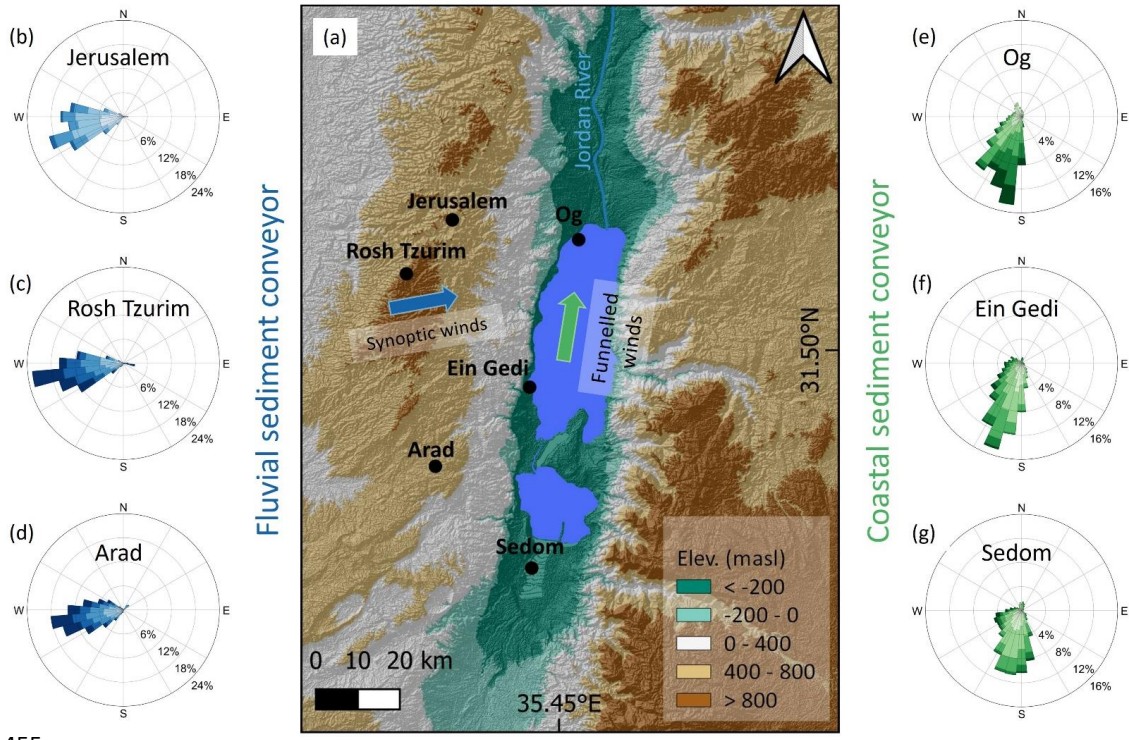

**Figure 10: Synoptic and meso-scale windstorms. (a) Location map showing the two perpendicular directions of the**
**winds flow during MC storms. (b, c, and d) Wind roses from three Judean Mountains water divide stations (locations**
**are indicated in the map). These data show the western-southwestern high-magnitude winds during winter storms**
**conveying at high altitudes the moisture for flood generation in the fluvial sediment conveyor (blue coloring). (e, f, and**
**g) Wind roses from inside the Dead Sea rift valley. These data show the change in wind direction as the synoptic scale**
**winds are funneled in the rift and transformed into high-magnitude southerlies that generate the northward**
**propagating storm waves activating the coastal sediment conveyor (green coloring). Legend of the wind roses appear**
**in Fig. 3c-d.**



### 4.3 The sediment conveyors at the seasonal scale under a joint atmospheric circulation generator

#### 4.3.1 The coastal conveyor at the seasonal scale

Like the stream, the coast is activated mainly between December and March (Fig. 11) under MCs located north of the Dead Sea region (Fig. 12). Each of the 128 classified storm waves (i.e., 10–30 storms per winter) are wind driven and are correlated with high magnitude southern winds (Fig. S6). The wind and wave storm durations are very similar or equal (Fig. 12a), ranging between several hours to three days, <1.5 days for the 25-75 percentiles of the wind (Fig. 13a-b). The prevailing CP during 80% of the identified storms is MC (Fig. 12a), also causing the highest storm wave energy with the longest duration of up to 3.5 days (Fig. S5). At the onset of storms, on average, a deep low-pressure system, ~10 hPa below mean, is located in the vicinity of either Cyprus or Syria, exhibited in the composite analysis as bi-center lows in these two regions, and the regional wind direction is western, with a slight southern component over southern Israel (Fig. 13d). At storm waves peak, the low-pressure system contracts its area and moves eastwards (Fig. 13e). Along the Dead Sea, the median wind speed at the storms peak is of 10 m s$^{-1}$ with short-term winds of up to <20 m s$^{-1}$ and a clear southern direction. The wind-driven northwards propagating waves, typically lagging the regional wind peaks by 0.5-2 h. Median wave height is about ~1 m with maximal height of ~4 m. The cessation of storms is associated with significant shallowing of the MC, appearance of high-pressure system and its advancement from the west, and a change of the mean wind direction into northwesterly winds (Fig. 13f), funneled inside the Dead Sea valley into northerlies.

The non-MC storm waves are generated by low wave-energy CPs, mainly by Active Red Sea Troughs, (15% of storm-waves producing CPs). The other 5% are caused by Persian Troughs and Sharav Lows, generating storms lasting <10 h (Fig. 12a, Fig. S4). Practically, these storms have a minor impact on the coastal geomorphology and sediment transport as the thresholds for the motion of clasts in the coastal conveyor are barely exceeded.

The comparison of the mean climatology of wind-wave producing MCs with the nonproducing MCs, show that wind-wave producing MCs: (i) are characterized by stronger regional westerlies, and (ii) have ~3 hPa deeper low center and an adjacent high of ~5 hPa higher located over Egypt and Turkey. This difference of ~8 hPa results in steeper pressure gradients from the north and south of the MC and the generation of stronger winds (Fig. 14), which are then funneled into southerlies at the local scale (Fig. 10).

#### 4.3.2 The fluvial conveyor at the seasonal scale

Flood-producing rainstorms in the stream occurred 4-9 times per season. Each of these rainstorms lasted between a few hours and up to two days (Figs. 11, 12b) with a typical duration of 10-15 hours for the 25-75 percentiles (Fig. 13c). These rainstorms have a median peak intensity of 5 mm h$^{-1}$ for the duration of an hour (Fig. 13c), and maximal intensities <20 mm h$^{-1}$ (Fig. 11). Rain depth >10 mm per storm generates moderate or larger floods as measured in the center of the Og watershed (Fig. S7). L 60% of the floods present low discharge with a peak discharge <10 m$^3$s$^{-1}$ or attenuate to such low flows that the floods do not reach the lake. Moderate floods (9 floods, 28%) experience peak discharge of 10–60 m$^3$s$^{-1}$ and the high-discharge floods (4 floods, 12%) have an estimated peak discharge of 60–170 m$^3$s$^{-1}$. Under rare conditions extreme floods with a peak discharge >170 m$^3$s$^{-1}$ can be generated. For example, in 2006, an exceptional discharge of 330 m$^3$s$^{-1}$ (Arbel et al., 2009), equivalent to an



instantaneous rainfall intensity of 8.7 mm h$^{-1}$ over the entire watershed, has been indirectly estimated in Nahal Og
based on high-water marks.
Approximately 85% of the flood-producing rainstorms were generated by MCs, with all the moderate to large
floods generated by this CP type. Moreover, these rainstorms occurred coevally with storm waves occurring under
the same MCs (Fig. 11). For MCs, rainfall amounts increase with storm duration (Fig. 12b), a relation that we
attribute to the characteristically continuous, wide coverage of rainfall during MCs (Armon et al., 2018). The
finding is coherent with similar analysis that was applied for the adjacent and larger Lower Jordan River (Armon
et al., 2019).
The rest of the flood-producing rainstorms (~15%) are attributed to ARSTs (Fig. 12b). These storms produced
low floods during the beginning and end of the hydrological season. This observation emphasizes the control of
MCs on geomorphic processes and delivery of sediments to the basin in this region (Fig. 12). For ARSTs, both
rainstorm duration and floods occurrence are uncorrelated with rainfall amounts (Fig. 12b); these complex
relations are attributed to the short duration and relatively high-intensity, localized rainfall associated with ARSTs
that a single rain gauge (Ma'ale Adumim, location in Fig. 2b) cannot capture, biasing the flood-producing rain
depth (e.g., Sharon, 1972; Marra and Morin, 2018).





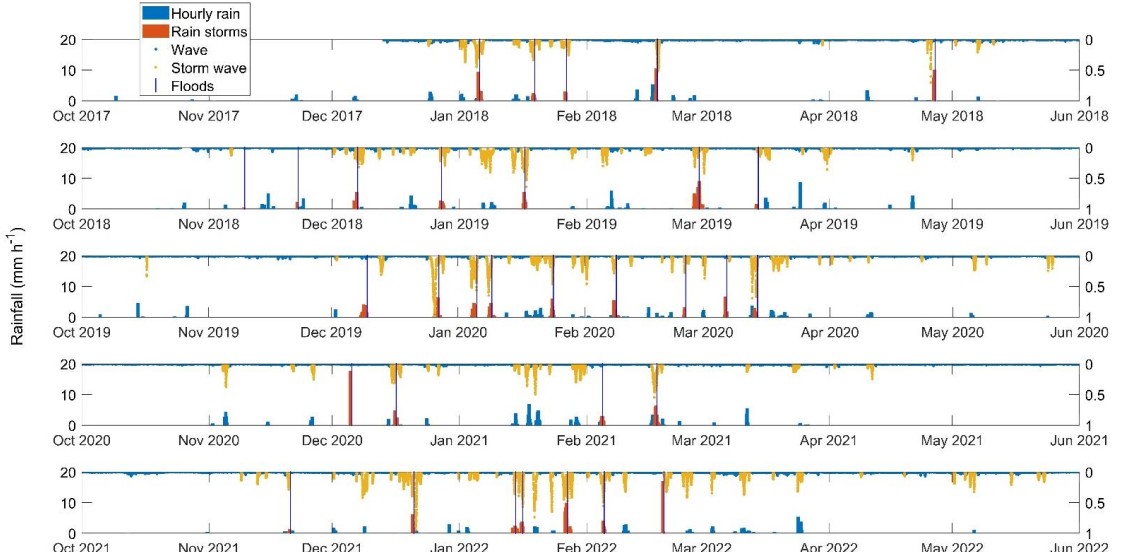

**Figure 11: The interaction between fluvial and coastal conveyors during five consecutive hydrological years 2017-2022.**
**Hourly rain depth measured in Ma'ale Adumim (location in Fig. 2b) with classified flood-producing rainstorms (left**
**axis; blue and orange bars, respectively). Vertical blue lines represent the occurrence of floods (Table S1). Waves with**
**classified storm waves (reversed, right-axis; blue and yellow dots, respectively).**

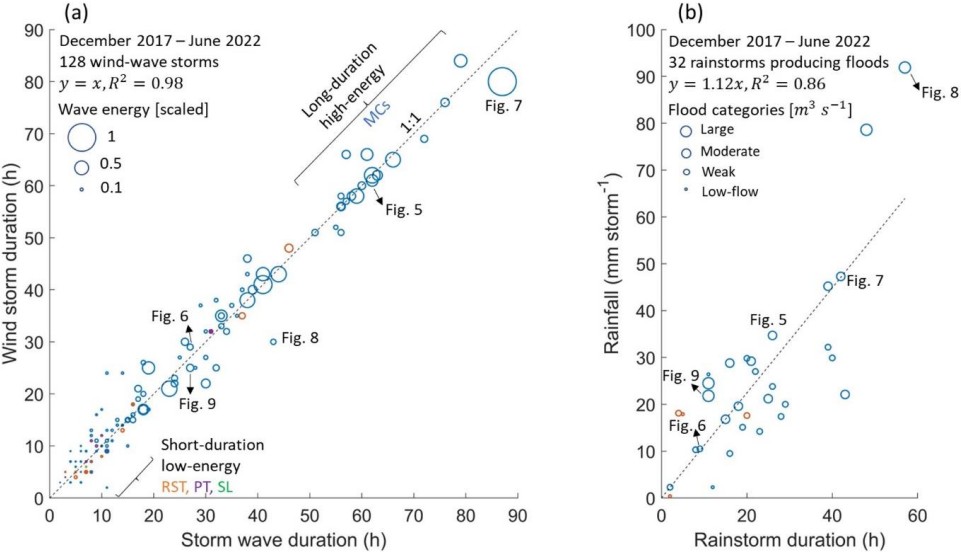

**Figure 12: (a) Duration of wind versus wave storms (circles), the energy of a storm wave (circle size), and atmospheric**
**CPs (MC-blue, RST-orange, PT-purple, SL-green). Storm wave energy was calculated for each storm according to**
$E \sim \sum H_{m0}^2$, **and then scaled between 0 to 1 according to the full range of storm wave energies. (b) Rainfall depth versus**
**rainstorm duration at rainstorms-producing floods (circles), the categories of floods (circle sizes), and CPs according**
**to the same color coding as in (a).**

Earth **Surface**
**Dynamics**
Discussions

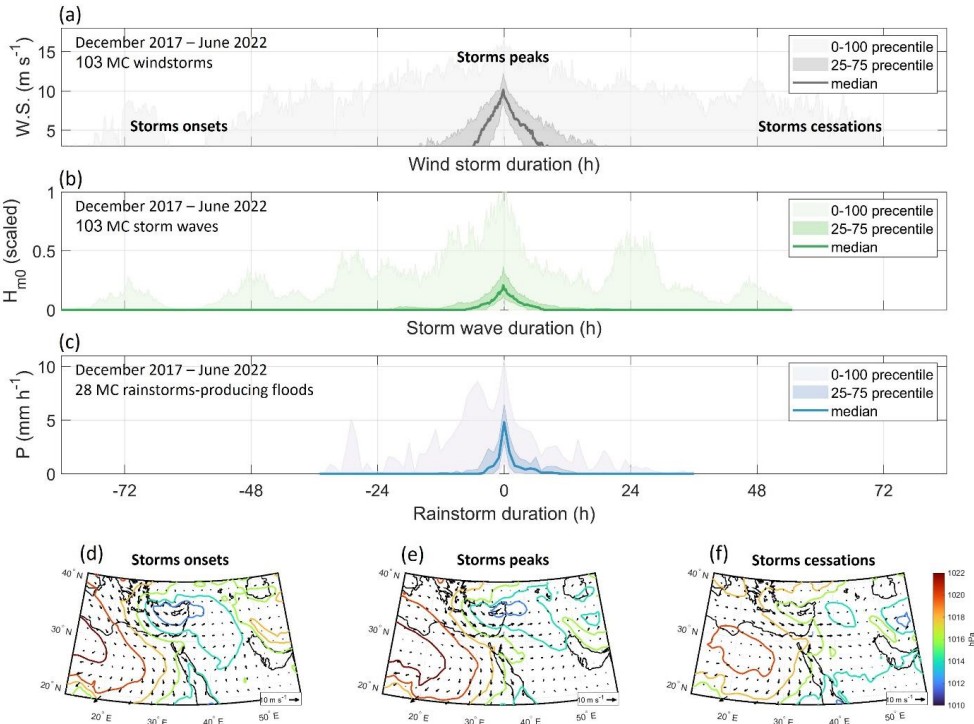

**Figure 13:** The 'mean' (a) wind speed, (b) wave height, and (c) flood-producing rainstorms under MCs. Median storms values (solid lines), intermediate quantiles of the storms (25-75%) and the full range of values (0-100%) is indicated (shaded-colored areas). Composite mean pressure maps at the (d) onset, (e) peak, and (f) cessation of the wind-wave storms showing the mean synoptic-scale evolution/climatology during the storms.

Earth **Surface**
**Dynamics**
Discussions

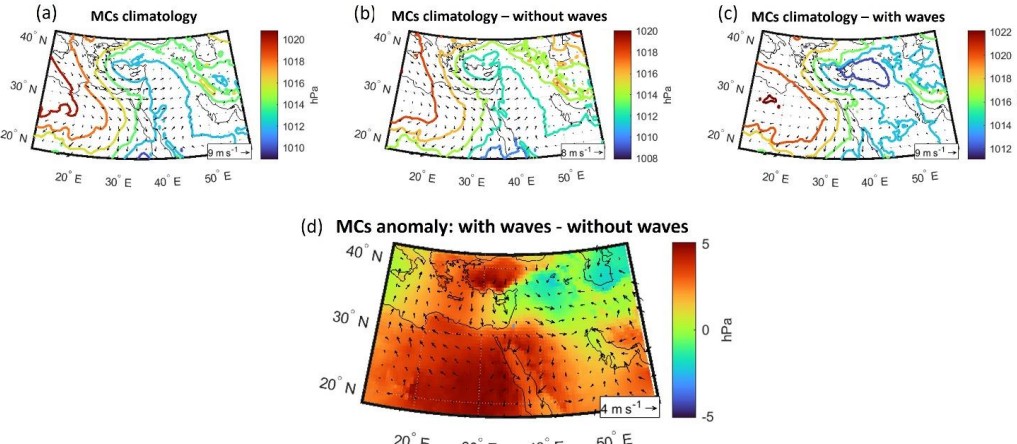

**Figure 14: The climatology and anomaly of MCs producing and non-producing wind-wave storms. MCs climatology composite pressure maps of (a) all days classified as MC (following Alpert et al., 2004), (b) the non-generating wind-wave storms, (c) the generating wind-wave storms. (d) The difference (subtraction) between the generating and non-generating MCs.**

## 5. Hydroclimatic signature in modern to paleo-sedimentary records

Following the detailed observations of waves, floods, and related sediment transport under MCs (Sect. 4), we discuss here the accumulation and architecture of modern and paleo-Dead Sea coastal landforms formed over longer time scales of decades to millennia. In Sect. 5.1, we discuss the accumulation of the Nahal Og modern-recent environment evolving across the Dead Sea shelf and slope under rapid lake-level fall of the past decades. Then, in Sect. 5.2, we present observations of a nearby stream and its coastal landforms accumulated on top of the shelf during the last modern Dead Sea highstand. Finally, in Sect. 5.3, we extend the discussion to gain insights into the architecture of fan deltas and paleo-beach berms formed during the Late Pleistocene at the foot of the Dead Sea western escarpment.

### 5.1 The evolution of Modern lowstand coastal berms (Nahal Og)

The sedimentary record of coarse-clastic beach berms at the Nahal Og mouth has accumulated since the early 2000s (Eyal et al., 2019) (Fig. 2d), pointing to three clear sedimentary/architectural trends over time: (i) Northward deposition of beach berms that (ii) lengthen with time under waves action, and (iii) increased sediment volume delivered by the incising and steepening stream to the receding shoreline (Fig. 15). The northward orientation of deposition is attributed to MCs-generated winter storms and northward propagating waves. However, the latter two trends contrast the hydroclimatic forcing of winter rain-floods and wind-waves that do not exhibit a significant trend in the past decades (Sect. S2). If anything, there may be a regional drying trend due to the poleward shift of the storm track and a decrease in total storm rainfall (e.g., Shohami et al., 2011; Zittis et al., 2022; Zappa et al., 2015; Hochman et al., 2018; Armon et al., 2022).

Therefore, the intensified sediment delivery to the basin is attributed to the geometric response of the channel to lake-level fall. Following the exposure of the Dead Sea shelf and steeper channel mouth gradients (~10%), a rapid



incision across the shelf was triggered (Eyal et al., 2019). An expanding knickzone evolved with higher gradients
migrating upstream (Ben Moshe et al., 2008), concurrently with channel deepening and narrowing that should
increase fluid shear stress exerted on the channel bed and bedload sediment flux to the channel mouth (Meyer-
Peter and Müller, 1948). Indeed, the transport rate across the shelf for a specific clast size increased over time
from tens to hundreds of meters per year over ~15 years (see discussion regarding the 'virtual velocity' in Eyal et
al., 2019). In larger spatio-temporal scales, it was shown that channel gradient is a first-order control on sediment
supply to river mouths together with the contributing drainage area (Syvitski and Milliman, 2007). The latter
factor is dominant along the global ocean shores during glacial periods when global sea level falls and watersheds
may merge over the exposed continental shelf (Mulder and Syvitski, 1996; BURGESS and HOVIUS, 1998),
supplying larger volumes of sediment into a certain lowstand delta (e.g., Anderson et al., 2016, for the rivers
draining into the Gulf of Mexico). The contribution of climate change during glacial lowstands is considered a
second order influencer (Syvitski and Milliman, 2007), with complex relations that may result in either increase
or decrease of the sediment delivery to channel mouths (e.g., Blum and Hattier-Womack, 2009) mainly of the
suspended sediment fraction (e.g., Mulder and Syvitski, 1996; Fagherazzi et al., 2004).
The lengthening of beach berms with time under annually similar wave climate is a less clear phenomenon as it
was concluded before that a single clast of a certain mass would travel a fixed, predictable distance under a given
distribution of wave heights within a storm (Eyal et al., 2021). This raises the question: why would larger sediment
volumes travel farther along the shore under a similar wave climate? Three mechanisms may explain this
observation: (i) The decay of wave orbital velocities with water depth (e.g., Dean and Dalrymple, 1991) results in
higher near-surface orbital velocities encountering large, thicker sediment volumes. Thus, the potential of gravels
to travel longer distances along the shore is higher for larger sediment volume. (ii) The probability of a clast to be
washed out of the swash zone during a storm coevally to the dominating stormy longshore transport (e.g., Benelli
et al., 2012). Lighter/smaller clasts have a higher probability to be washed out of the swash zone than
heavier/larger clasts that tend to travel down the beach slope under the influence of gravity (e.g., Grottoli et al.,
2015). Consequently, smaller sediment volumes, characterized by smaller grain-size distributions (Eyal et al.,
2019), have a higher probability to completely be washed out of the swash zone at early stages of the season,
forming shorter-extending beach berms. (iii) Reworking of beach berms between successive years. Lake-level
declines at ~1.2 m y$^{-1}$ over the relatively steep (~10%) beach slope, exposing annually 10-15-m wide strip of the
previous year coastal sediment, leaving <50% of the coarse sediment submerged underwater. This way, sediments
that have travelled along the shore in the previous year, start moving from an 'advanced' location, and reach
farther northward distances. This inter- annual process is superimposed on the existing signal of increasing
sediment volumes conveyed to the coast with time. Gravels weighing several kilograms travel distances of
hundreds of meters during single storms between 2018-2022 (Figs. 5f, 9f), an order of magnitude longer distance
than the shortest beach berm preserved in the Nahal Og from the early 2000s with a length of tens of meters (Fig.
2d). This observation strengthens the assertion that for larger volumes of sediment, gravels are displaced farther
along the shore, and the inter-annual recycling between beach berms, may be superimposed on the signal of beach
berms lengthening with time.

Earth **Surface**
Dynamics
Discussions

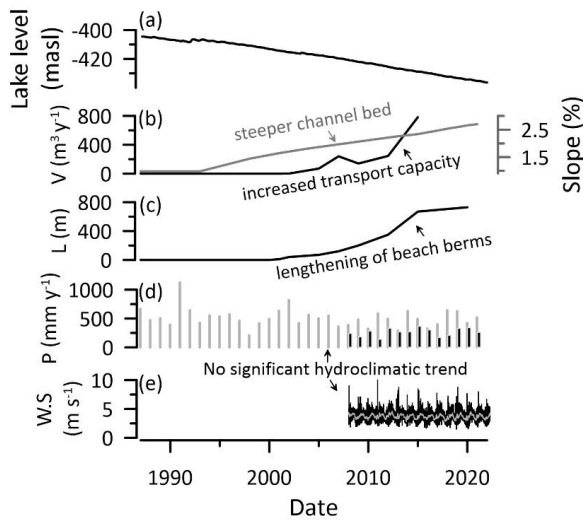

**Figure 15: Reorganization and the buildup of lowstand sedimentary record under hydroclimatic forcing. (a) Dead Sea**
**lake level. (b) Average channel slope of Nahal Og, measured between Highway 90 to the Dead Sea (Fig. 2c), increase**
**with time in response to rapid level decline (right axis; grey), the estimated increase in annual volume (V) of sediment**
**delivered to the channel mouth following Eyal et al., (2019) (left axis; black). (c) Increase in the length (L) of beach**
**berms with time. (d) Annual rainfall (P) in Ma'ale Adumim (black bars, 2008-2022) and Jerusalem (grey bars, 1985-**
**2022). (e) Wind speed (W.S) in Beit Ha'Arava (black line; daily mean, grey line; monthly mean, 2008-2022).**

**5.2 Modern highstand coastal landforms of a nearby stream (Nahal Qumeran)**
The northward elongation of beach berms deposited during the highstand phase of the early 20th century Dead Sea
at the mouth of a nearby ephemeral stream, Nahal Qumeran (Fig. 16a-c) provides a wider perspective of our
analysis. The Nahal Qumeran catchment neighbors Nahal Og from the south (Fig. 2b,c), it has a smaller (47 km$^2$)
and drier mean annual rain volume over its watershed of $8 \times 10^6$ m$^3$ y$^{-1}$ (Ben Moshe et al., 2008) than Nahal Og.
Between 1945 to 1960 the Dead Sea level was relatively stable, ranging between -390 to -395 mbsl, and Nahal
Qumeran was fluvially connected to the Dead Sea shores through a braided coarse-clastic fan-delta. During the
1960s and 1970s, with the onset of human-induced lake-level decline, the stream could keep pace with the slowly
regressive shoreline to feed its highstand fan-delta (Fig. 16b,c). During this interval, a series of beach berms
similar to those formed in Nahal Og, were formed, showing extension to the north from the Qumeran channel
mouth, fitting the above detected preferred directionality of winter winds and storm waves (Sect. 4). We do not
identify any trends of increased sediment volumes or lengthening of beach berms in the channel mouth as its base
level is approximately stable and the channel profile and sediment flux are not interrupted. Since the early 1970s
lake-level decline has accelerated, the channel did not keep pace with the rapid receding shoreline and low-
gradient mudflats emerged (see also Eyal et al., 2019; Enzel et al., 2022). At that moment, Nahal Qumeran stopped
responding to the rapid lake-level decline and became disconnected from the lake, showing no incision across the
shelf or any sediment delivery to the lake (Eyal et al., 2019). Instead, this stream maintains the buildup of an
alluvial fan prograding onto the mudflat platform, with no substantial impacts of the lake coastal hydrodynamics





that generate the northward depositional asymmetry, related to the regional forcing of MCs. It seems that as long
as the fluvial and coastal conveyors interact, regional hydroclimatology was manifested in northward elongating
beach berms, similar to Nahal Og. However, disconnecting the fluvial from the coastal conveyors, transforms the
channel mouth from a fan-delta into an alluvial fan that develops onto the mudflats regardless of the water body
hydrodynamics.

**5.3 Pleistocene Lake Lisan - sedimentary record of Nahal Tmarim**
Following the observations from the modern Dead Sea in Nahal Og and Nahal Qumeran, we explore whether the
control of southern winds along the Dead Sea rift valley, had affected past deltaic-coastal sedimentary
morphology. At the foot of the western Dead Sea escarpment at stream outlets, Gilbert-type fan-deltas, alluvial
fans, and paleo-shorelines associated with the higher levels and recession of the Late Pleistocene Lake Lisan are
well-preserved (Fig. 16a,d,e; see Fig 2b for the extent of Lake Lisan) (e.g., Manspeizer, 1985; Frostick and Reid,
1989; Bowman, 1971, 2019). We have recognized an asymmetry in the deposition of fan-deltas along most of the
northwestern shores of the Dead Sea in both large and small streams; they present preferential deposition and
more pronounced shorelines north of the feeding canyon mouths (Sect. S7). Channel outlets from the Deas Sea
escarpment/cliff maintain their locations since the Late Pleistocene as successions of Lake Lisan deposits are
preserved inside deeply incised canyons at stream banks (e.g., Bartov et al., 2007). Thus, the depositional
geometry and asymmetry of the channel deposits are evaluated with respect to the channel outlet from the Dead
Sea escarpment as an indicator of their deposition due to funneled wind and wave storm direction in the Late
Pleistocene. Here we present an example from the outlet of Nahal Tmarim (~22 km$^2$ drainage area), located ~15
km south of Nahal Og (Fig. 2b,c). Its Pleistocene fan-delta and its recessional paleo-shorelines/beach berms are
deposited at elevations ranging between 310 to 330 mbsl, corresponding to lake level decline of the Late
Pleistocene to Holocene (e.g., Bartov et al., 2007). The depositional configuration shows the abovementioned
asymmetry, with most of the sediment volume of the fan-delta extends northward of the stream outlet from the
cliff (Fig. 16d,e); the surface area of deposits north of the channel outlet is four times larger than the depositional
area south of the outlet. Furthermore, sorting of cobbles-boulders is observed along the paleo-shorelines; clast
sizes decrease northward and away from the Tmarim channel outlet, whereas, practically, no shorelines/berms are
recognized south of the stream outlet. The fan-delta of current Nahal Tmarim is different from the modern fan-
deltas of Nahal Og and Nahal Qumeran in several aspects: (i) It is a thick (20-30 m) deposit with Gilbert-type
forests and paleo-shorelines preserved on its surface. (ii) There is some contribution of coarse materials to the
coastal system either directly through the cliff taluses or by debris flows occurring under exceptionally heavy
storms (David-Novak et al., 2004; Ahlborn et al., 2018). (iii) It was built during Lake Lisan highstand and got its
final geomorphic shape during the regression of the lake (27-14 ka ago) and the transition into the Holocene
conditions, 14-12 ka ago (e.g., Bowman, 2019). Despite these dissimilarities, the framework under which this
sedimentary record had evolved with the northward extension of the delta, is similar. It indicates a dominating
southern wind-wave regime and a signature on past sedimentary records during the latest Pleistocene, were very
similar to today.
The highest stand of Lake Lisan ca. 26,000 years ago reached 145-165 mbsl (Bowman and Gross, 1992; Bartov
et al., 2002; Abu Ghazleh and Kempe, 2009), and extended over 240 km, from the Sea of Galilee to the northern
Arava (e.g., Bartov 2007) (Fig. 2a). The potential length of the fetch at what is currently the northern Dead Sea



basin more than doubled from both the north and the south. Thus, both northerlies, presently driven by meso-scale
circulation of Mediterranean Sea breeze (e.g., Lensky et al., 2018), and southerlies, mainly driven by synoptic-
scale MCs, could have potentially generated waves high enough to transport gravels along the shores of the lake
in both directions. However, the observed preferential deposition asymmetry points to the southerlies control and
in turn, to MCs that generated these southerlies-driven-waves with transport of coarse gravels northward; there is
no evidence for a preferred fetch from the north.
Moreover, the northward directional organization of coarse sediments in the basin agrees with the increased
frequency of MCs during wetter intervals of high lake stands in the Dead Sea basin (Armon et al., 2019; Enzel et
al., 2008; Ben Dor et al., 2018). This inference is based on present-day climatology showing that wetter winters
and high-lake levels are characterized by higher frequencies of deeper and southerly displaced storm tracks of
MCs (e.g., Ben Dor et al., 2018; Enzel et al., 2008, 2003; Saaroni et al., 2010). Prevalence of more frequent,
deeper MCs during the wetter Late Pleistocene, should have been resulted in an intensified activation of the *fluvial*
and *coastal sediment conveyors,* compared with modern conditions, as MC is the only CP that can generate both
rainstorms and windstorms in this region. Floods were more intense and probably more frequent, they have
delivered amplified sediment fluxes into the basin (Bartov et al., 2007). Westerlies/southwesterlies funneled in
the rift valley into southerlies were more frequent and intensified, blowing over a longer lake fetch of
diluted/fresher and less dense water, thus generating higher waves, with maximum heights that exceeded the
modern 4 m. Such waves are characterized by higher fluid orbital velocities that generate higher forces to transport
larger boulders for longer distances along the coast.

Earth **Surface**
Dynamics
Discussions

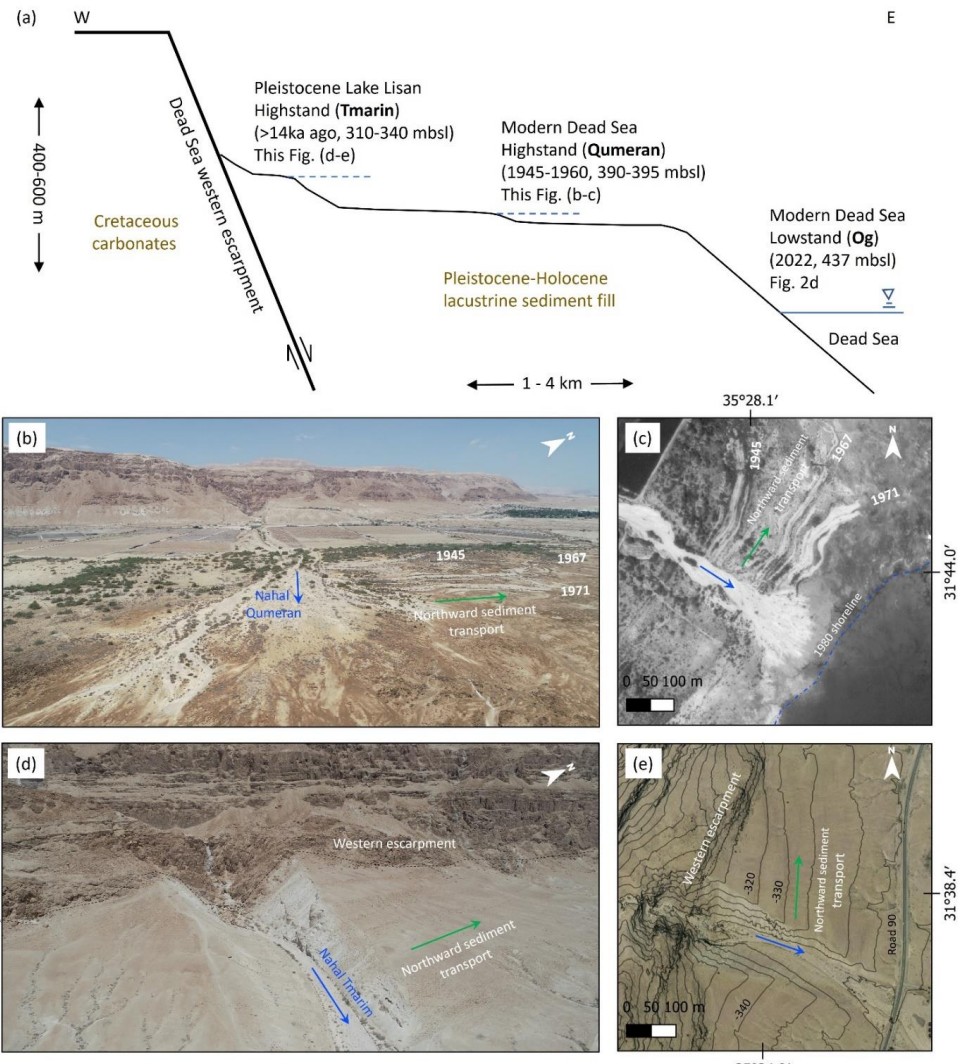

**Figure 16: Modern and paleo-northward-extending beach berms and fan deltas. (a)** Schematic cross section from the western Dead Sea escarpment to the modern Dead Sea showing the stratigraphic/geomorphic location of the three geomorphic records discussed in the paper. For location of the sites see Fig. 2b-c. **(b)** Angular drone photograph of Nahal Qumeran, and **(c)** orthophoto of Nahal Qumeran (1980), both showing the northward extending beach berms deposited as long as the stream fed the earlier 20[th] century shorelines with sediments. Since lake level decline has accelerated, the stream did not keep pace with the receding shore and an alluvial fan begun developing on top of the exposed shelf. **(d)** Angular drone photograph of Nahal Tmarim, and **(e)** orthophoto of Nahal Tmarim (2012), both showing the norward deposition of fan-delta and beach berms under late Pleistocene Lake Lisan wind-wave regime. The asymmetry of sediment deposition to the north is evident also by looking at the elevation contours in (e), converging with steps of pleo-shorelines, with respect to the escarpment strike; northward of the channel, contours are sub-parallel to the escarpment direction, whereas they diagonally approach it on the southern part.





## 6. Summary and conclusions

Mediterranean cyclones are the main synoptic-scale generators of both rain and storm waves over the Dead Sea region. Thus, they are also the main drivers for the coarse-clastic fluvial sediment flux into the lake and the transport and sorting of clasts along shores. First, these MCs generate the high-magnitude synoptic wind with westerly cyclonic circulation propagating to the northeastern Mediterranean. Near the surface and perpendicular to this synoptic wind direction, the flow is funneled orographically along the Dead Sea rift valley into southerlies that generate waves activating the *coastal conveyor*. Then, when the cyclone position migrates closer to the eastern Mediterranean shoreline or is centered inland in Syria, the northern component of the wind becomes more prominent, the southerly wave-producing winds decay, and rainfall evolves in the watershed over the Judean Desert. The rainfall generates floods, which activate the *fluvial conveyor* within a few hours. Thus, fluvial sediments reach the basin either coevally with or completely after the decay of the storm waves. Accordingly, the longshore transport and sorting often occurs during the next storm in the same season, or infrequently, over the same cyclonic system.

MCs-producing waves are, on average, ~10 hPa deeper, generating southern winds of up to 20 m s$^{-1}$ that last >10 hours. When the wind-driven waves are higher than 0.6 m, the threshold for transport of a 1-kg clast, the coastal conveyor is activated. When rainfall of >10 mm per storm accumulates at the center of the watershed, moderate flood or larger are likely to activate the fluvial conveyor.

Although both the stream and coast are usually activated under MCs, the transport under storm waves is >five times more frequent than the delivery of sediments by moderate or larger floods. This is geomorphologically noticeable in the wave-dominated fan-delta, transformed into regressive beach berms extending northward of the Nahal Og mouth. As the flood hydroclimatology shows no clear trend in recent decades, the increase of sediment volume delivered to the channel mouth during this interval, is attributed to the response of the stream profile to base level fall, the exposed stream mouth is steep and result in incising, steepening, and in increased bedload transport capacity. Concurrently, under rather constant wave climate, this increase in sediment discharge is associated with longer transportation distances of coarse gravels along the shore, and the increase of the beach berms length with time.

Guided by the observation from modern environments, we recognized that similar directionality of the hydroclimatology resulted in sedimentary deposition northward of canyon mouths in fan-deltas and coastal deposits from the Late Pleistocene. This implies that over past millennia, MCs have played major role in connecting fluvial delivery of coarse sediments, and their distribution in the lake and along its coasts.

## 7. Data availability

The data related to this work is available on Mendeley Data repository https://data.mendeley.com/drafts/65bhpwftrh (Eyal et al., 2022), and in Table S1 in the supplement. Rain gauge data were provided and pre-processed by the Israel Meteorological Service (https://ims.data.gov.il/; they are freely available in Hebrew only). ERA5 data can be downloaded from https://cds.climate.copernicus.eu (Hersbach et al., 2020). Flood reports from the years 2019-2022 were obtained from the Desert Floods Research Center (https://floods.org.il/english/; they are freely available in Hebrew only).



## 8. Video supplement

The videos related to this article are available on https://photos.app.goo.gl/rLysYEfoVSzyGdQo7.

## 9. Supplement link

## 10. Author contribution

HE, MA, and NGL conceptualized this work. The methodology was developed by HE, MA, and NGL. Data curation and formal analyses were performed by HE and MA. Funding was acquired by NGL, YE, and HE. NGL and YE supervised the work. HE wrote the original draft of this paper, which was reviewed and edited by all authors.

## 11. Competing interests

The authors declare that they have no conflict of interest.

## 12. Acknowledgements

This study was funded by the following grants: PI-NGL: ISF-1471/18, BSF-2018/035, NSF-BSF-2019/637; PI-YE: ISF-946/18. HE is grateful to the Azrieli Foundation for the Azrieli Fellowship. MA was supported by an ETH Zürich Postdoctoral Fellowship (Project No. 21-1 FEL-67), by the Stiftung für naturwissenschaftliche und technische Forschung and the ETH Zürich Foundation. We thank Vladimir Lyakhovsky, Eckart Meiburg, Efrat Morin and Itai Haviv for discussions and insights. We acknowledge Ziv Mor, Ido Sirota, Raanan Bodzin, Uri Malik and Hallel Lutzky for the assistance in the field and laboratory and Liran Ben Moshe for the drone photography. Dorita Rostkier-edelstein and Lida Shendrik are acknowledged for providing the updated synoptic classifications following Alpert et al., 2004, and Yoav Levi for sharing the large IMS datasets of rain and wind.



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
