# Peer review of "Synoptic- to meso-scale circulation connects fluvial and"

_Earth Surface Dynamics, 2022_

## Author Comment (AC1)

Below, we present reviewer's #1 comments and our respective responses in red text.

This manuscript presents hydrological, meteorological, and sedimentological data from over a five-year timespan along the west coast of the Dead Sea. In many ways it resembles a "source-to-sink" type study of two decades earlier: it traces the hydrometeorological conditions under which sediment can be exhumed from the landscape and delivered to the coast, where it is reworked alongshore by those same or subsequent storm events. One key aspect that allows this manuscript to stand out from those earlier efforts is that this study focuses on an interior basin with rapidly falling water levels; thus the coastal depocenter is one marked by forced regression and attendant preservation of falling-stage deposits (largely in the form of coarse to mixed clastic beach ridges), rather than a prograding delta or deep-water marine environment.

The manuscript is thorough and data-rich. It is structured well and provides adequate background and information, particular given how heavily it relies on previous work by these authors and others (all well-cited). Figures are generally very clear, well-constructed and add to the text by providing detail and context through examples from individual events (I provide some suggestions below for minor edits).

Thanks for this positive feedback.

The manuscript does, however, require some careful proofreading and editing: there are commas out place, missing words, noun-verb disagreement, inappropriate word choices, random capitalization, etc. These are all minor and do not detract from the science or paper, but readability would be improved if fixed. I have provided several examples in the "detailed" edits given below.

We carefully reviewed and proof-edited the manuscript according to your comment. It can be seen in the attached 'compared versions' document.

Along with this, I would suggest that the authors reduce the use of acronyms as possible, and be sure to define all of them in use prior to first use (e.g., "ARST" is used on L172 but I believe first defined on L296, along with other acronyms which had already been defined).

We implemented this suggestion and now all acronyms are defined prior to their first use.

The science presented appears sound and conclusions well-founded. Comparisons in the Discussion to published works from nearby river mouths extend and strengthen the inferences made here. However, section 5 and 6 are missing a discussion of the broader implications of this work. The role of storm-associated flood events in delivering sediment (including coarse gravels) to the coast, and of high wave energy events transporting those alongshore (particularly along wave-dominated coasts), is well known. Thus, as is, this remains somewhat of a site-specific study: certainly highly appropriate for publication, especially given that this is a relatively understudied region of the world (as are ephemeral fluvial systems in general). But, additional analysis placing these findings in the context or other river and coast dispersal systems would expand the likely audience.

We extended the discussion in section 5 using studies by Nienhuis et al. Now, the findings from Nahal Og are presented in the context of predicted plan-view evolution of

wave dominated deltas. We highlight the role of lake level fall and absence of updrift littoral transport on the architecture of a lowstand delta\ receding shorelines.

Finally, I note one location where some additional analyses might strengthen the manuscript: in section 5.1, the authors discuss coastal change in terms of the length of beach ridges (berms), and conclude that these record an acceleration in sediment delivery to the basin attributed to enhanced incising due to base-level fall. The conclusions and inferences are sound, but the manuscript contains no evidence of quantification of these changes in sediment delivery. Figure 2d shows the locations of recent paleoshorelines. The authors might consider trying to calculate sediment volume fluxes into these shorelines through time. Subsurface data (ground penetrating radar would likely work well in this environment) could be used to map deposit thickness through time to estimate volumes and properly demonstrate the increase in sediment load. Such an approach would also consider shoreline orientation and thus plan-view estimates of shoreline area built over a given period of time.

Estimate of sediment volume fluxes ($m^3y^{-1}$) through time and the increase in sediment load appear in figure 15b with data based on Eyal et al., 2019. We added more accurate referencing to the figure in the first paragraphs of sect.5.1. We also added more information about the field setup that enables the quantification of sediment volume fluxes in the regional setting, Sect.2.1. Abandoned shorelines at a certain elevation are correlated with time through the well-established lake level curve. The coarse clastic bright material comprising these shorelines is clearly distinguished from the underlying dark silt-clay laminated lacustrine deposits (Fig. 2d). Thus, the length, thickness and width of individual berms is measurable in the field and its volume is approximated with a triangular-pyramid geometry (Eyal et al., 2019). These are direct observations of how the coarse sediment volume flux have evolved with time.

The thickness of the coastal deposits is measurable in outcrops exposed along the channel bank. Thus, we think that ground penetrating radar (GPR) will not add a lot of information to what we have already directly extracted in the field. Furthermore, preforming GPR experiments along the Dead Sea shores is a challenge, as the area is saturated with a dense brine which is highly conductive dispersing the radio energy.

Moreover, in Lines 585-588, the authors posit that large gravel travel further in more recent years because there is simply more large gravel to move. But, the coast is building out as base level in the Dead Sea falls. So, does shallow nearshore bathymetry play any role here? If the nearshore bathymetry steepens offshore, then wave energy reaching the beach would be expected to increase, thereby increasing the rate of sediment transport and the size of clasts transported. These types of three-dimensional considerations are critical to understanding the processes shaping the coast, and thus be able to relate that to sediment inputs (and upstream incising). These are completely lacking at present. At a minimum, producing a map of progradation rates through time (and assuming progradation of ridges with similar thickness onto a planar antecedent surface) could help to quantify the (assumed?) increase in sediment delivery. This would also fill a substantial gap in the work, in that any depocenter geomorphic analyses are lacking.

Sect. 5.1 was reorganized to clarify our statements as follows:

    (1) In the third paragraph of sect. 5.1 we discus mechanisms and reasons to explain the lengthening of beach berms with time. We suggest that under constant wave

climate, the main reason for the lengthening should stem from the increase in sediment volume flux due to the steepening channel (first and second mechanisms). We now clarify how larger sediment volumes are transported to longer distances alongshore, by presenting explicitly a direct field observation: "During the early 2000s, when small sediment volumes were delivered to the shore, beach berms of <100 m were formed (Fig. 2d, Fig. 15c), whereas between 2018-2022, larger sediment volumes were delivered to the shore and gravels were displaced longer distances of hundreds of meters along the shore at single storms (Figs. 5f, 9f)." Following that, in the next part of section 5.1, we discuss the possible mechanisms of shoreline lengthening with time.

(2) We clarify the effect of nearshore bathymetry on the energy of waves arriving the shore. Our observations show that the gradient of the shore face exposed with time is relatively constant. The domination of waves does not allow here any depocenter to evolve, fluvially derived sediment are completely carried from the channel mouth downdrift. The full analysis of the exposed slopes can be seen in Eyal et al., 2019, Figs. 2d and 5c. We added in the second paragraph of sect. 5.1 a sentence emphasizing the relatively constant value of the exposed slope\bathymetry. The study area provides a simpler case in which the fluvial sediment contribution does not change the slope and shoreline orientation, allowing the arrangement of the fluvial sediments along shores with clear separation; this is now implemented in sections 2.1 and 5.1.

**Detailed Edits**

Below are largely examples for improvements to writing clarity and precision noted above. However, several are questions and recommendations from specific parts of the manuscript.

Lines 79-80: provide examples.

We reorganized the paragraph and now the advantages of the Dead Sea environment are presented alongside with related knowledge gaps and examples.

Line 97: "nature" is non-quantitative. What does this mean here?

The question was rewritten: 'What are the characteristics of atmospheric CPs during which the fluvial and coastal conveyors are activated?'

Line 139: comma before "are fossilized" seems unnecessary.

Corrected.

Line 167: what is the "heart" of the winter? Be precise.

We added to the sentence that the heart of the winter occurs between November to March.

Line 172: "ARSTs" is not previously defined.

Corrected. Now it is defined earlier in section 2.2.1.

Line 225-227: wordy and complex. Consider breaking into multiple sentences and overall shortening.

This phrase was shortened and broken into two sentences.

Line 265: how many is "tens"? Be precise.

Corrected.

Lines 266-268: this is an incomplete sentence. There is no verb.

The sentence was corrected.

Lines 306-307: unclear writing and unclear how this conclusion was reached.

We clarified and rewritten the sentence.

Line 478: "lagging" should probably be "lag".

Corrected.

Line 496: what is "L" before "60%"?

The typo was corrected.

Line 562: why are some names in all caps?

The typo was corrected.

Line 582: from where does the estimate that <50% of coarse sediment is submerged underwater originate? How was this calculated? This is important as it comes back to the poor or missing (as described above) estimates of beach volumes that seem to feed directly into the conclusion of accelerated sediment delivery due to downcutting.

We now better explain this estimation in the text. The width (in cross shore direction) of the strip of gravels extending along the shore is about 20-30 m. Thus, under lake level decline of ~1.2 m $y^{-1}$ over a lakebed slope of ~10%, about half of the coarse gravels strip is exposed every year. From here stems the conclusion that <50% of the coarse sediments stay submerged underwater.

Line 583: why is "advanced" in quotes?

The quotes were removed.

Line 643: the name of the study site is misspelled.

The typo was corrected.

**Figures**

Figures are generally excellent. Some minor suggestions are:

New Figure: the authors might consider a new figure to supplement section 2.1.1. A map showing predominant wind and storm conditions (as is common in many climate and paleoclimate studies) would help illustrate this complex system for the reader.

Figure 10 summarize the regional, predominant windstorm conditions over a map as reflected in the related storm wind roses. Additionally, Figure 3b-c provide the continuous wind regime presented along a timeseries and by wind roses. In the supplement Sect. S2 we compare the five years-long winds to a longer wind record monitored in an adjacent wind station (Beit-HaArava).

Figure 2: the keys and some other text in (a) and (c) are illegible (dark text on dark blue background). Also, is it accurate to call these geomorphic features along an exposed lake shoreline a "shelf" and "slope"? Those are typically reserved for large-scale continental margin features within marine environments.

The contrast between the text and background was improved.

We added to the figure caption the sentence "We adopt along the Dead Sea margins the global terminology for shelf and slope because of their similar geometry (see Eyal et al., 2019)."

Line 383 (Figure 5): it is not clear what the gradient fills illustrate, or why they are gradients of blue and gray. (same comment stands for figures 6, 7, 8, and 9, and associated captions).

It is written in the captions of figures 5-9 that the colored gradient fill indicates waves above transport threshold. We now clarified that the winds and waves are plotted in black and blue, respectively, and added that the color fill darkens towards the higher values of both the winds and waves.

Figure 10: some dark text is illegible on dark background. Otherwise, a very nice figure that helps illustrate the different forcings within the drainage basins and along the Sea coast.

The contrast between the text and background was improved.

Figure 13: the light purple, green, and blue shading (0-100 percentile) is very difficult to see. Suggest darkening all.

Colors were darkened.

---

## Author Comment (AC2)

Below, we present reviewer's #2 comments and our respective responses in red text.

I reviewed the manuscript of Eyal et al., documenting coarse-grained fluvial and coastal sediment transport around river mouths in the Dead Sea basin, and their connection to atmospheric circulation patterns.

The paper presents new field data about a river delta in an arid environment subjected to base level fall -- conditions which I am not very familiar with. I found it interesting to read, but I noticed I had to go back to the maps a lot to better understand the local geography and the explanations in the text. A map showing the "berm" that is frequently discussed would be helpful.

A map showing the berms that are discussed along the manuscript, appears in Figure 2d; we added to the figure a label highlighting the berms and the silt-clay dark lacustrine sediments on which they are deposited. We improved the referencing to the figure and extended the explanation of the field setup in Sect. 2.1.

I have also some general feedback that I think would me (as a coastal geomorphologist) better understand the paper;

1) I was wondering about the connection between atmospheric circulation patterns and wave transport direction. It seems that, given the location of the delta within the Dead Sea, littoral transport to the North is inevitable and virtually independent of atmospheric circulation. Wave height is determined to a large extent by the fetch, and the only significant fetch is toward the south, which then drives transport to the North. I don't think any other weather types would be visible in the (paleo) record. This does not make it a very good place for paleoclimate reconstructions, but perhaps I am missing something here?

The meteorological analysis shows that both southerlies and northerlies dominate the wind regime along the Dead Sea rift (Figures 3c and S2). However, only the southerlies are of high magnitude and long durations during winter storms (Figure 10e-g), regardless of the fetch available in the study site along the Dead Sea. Thus, although under modern conditions northerlies cannot generate waves contributing to longshore sediment transport because of the limited fetch from the north, they are anyway of lower magnitudes and shorter duration and won't generate significant transport. The inspection of the geometry of abandoned modern and paleo records, strengthens our paleoclimatic conclusions as showing the prevalence of northward unidirectional transport of coarse gravel also during the last glacial period. Then, the lake was larger with a fetch of ~100 km north of the study site. See also the detailed discussion in the end of Sect. 5.3.

The unique configuration of the study site with forced regression, separation between annually formed beach berms, and distinction between the autochthonous muddy lacustrine deposits and allochthonous coarse gravel, enabled to study the modern active environment and then interpret similar hydroclimatic trends in the geomorphology of nearby paleo sediment records.

2) The argument in section 5.1, about coastal berms, I found difficult to understand. The way I read it, it seems to me that the boulder are a tracer of local river-derived sediment. But: the delta geomorphology in this case is a result of river-derived sediment as well as updrift coastal sediment supplied by the waves. This delta seems strongly wave-dominated, such that most of the alongshore transport is updrift rather than river-derived.

A figure could be helpful here to better understand the geography/time evolution of the berms, perhaps also with a mass balance to constrain the fluxes.

We clarified our statements in section 5.1. It is true that the delta is strongly wave dominated and sediment volumes arriving to the coast by the stream are completely transported downdrift alongshore by the waves. Specifically, we added that "coarse sediments are sourced only from the stream with no littoral updrift sediment contribution".

Sediment fluxes are presented in figure 15b and the geography with time evolution in figure 2d. We better reference these figures now. Furthermore, we improved the description of the field setup in Sect. 2.1 of the regional setting. The study site is unique and enables to directly estimate the annual volume flux of coarse sediments along abandoned beach berms. The stream is an independent source of coarse sediment to the delta and beach berms, according to the following observations: (1) south of the channel mouth there are no coarse-clastic materials and the lacustrine muddy lakebed is exposed, and (2) nearby gullies are local, draining the muddy areas of the shelf and are not connected with a drainage basin upstream, thus unable to transport coarse materials.

3) The relation between transport and orbital velocities is not very clear (around L568). Alongshore transport of sediment is commonly calculated with the CERC formula, or its equivalent. There is a dependence on wave height and wave approach angle. An increasing river flux (from a steepening channel bed) would change the shoreline orientation and thereby also the wave approach angle. An asymmetric wave climate would steer the delta in a downdrift direction (see a study of mine about this for some better explanations: Nienhuis et al., EPSL 2016).

Thanks for this comment. We extended the discussion in the end of Sect. 5.1 dealing with the processes described in Nienhuis et al., 2016 compared with the setup of the Dead Sea. The main differences are the effect of lake level fall and absence of updrift littoral transport in our study site. Although river flux increases with time, the shoreline orientation doesn't change as waves energy is sufficient to transport the annual flux of coarse sediment along the shore preventing delta grow up\progradation in the channel mouth.

Orbital velocities of the waves\breaking waves are mentioned in the discussion of 'why would annually increasing sediment volumes travel farther along the shore under a similar wave climate?'. We suggest that larger sediment volume accumulate up to shallower water depth and are subjected to higher near-surface wave\breaking-wave orbital velocities, relative to smaller sediment volumes on which lower fluid velocities are exerted at deeper water depth. Thus, the potential of gravels to travel longer distances along the shore is higher for larger sediment volumes.

Some small additional comments:

L19: "rise" and fall?

We prefer to stay with only 'fall' as in this manuscript we deal with the response of streams and coasts to Dead Sea continuous level fall, i.e., the case of geomorphic response to forced regression. Geomorphic responses to lake level rise are related to

less frequent rainy winters as we present and discuss in a recent paper by Enzel et al., 2022.

L23: "perpendicular" sounds odd to me. Up, down, left, right? Perhaps write cross-shore vs alongshore, or fluvial vs. alongcoast etc.

We rephrased the sentence.

L29: "dominates"-> dominate

Corrected.

L32: this is the first time you mention that you studied paleo records as well; perhaps include that in your list of methods in L19-20?

We try to be concise as much as possible in the abstract preferring to focus the list of methods to the reach dataset we collected in the past years. Then, we present the extension and implications of the analysis dealing with paleo records a few sentences later.

L40: "also"? as opposed to what?

'also' was omitted and the sentence was rephrased.

L43: "jointly" refers to basin and terrestrial controls?

'Jointly' refers to the fluvial and coastal conveyers. The paragraph was rephrased. And now it should be clearer.

L58: there is a lot of literature on beach change and climate signals (nao, enso) so I'd be careful with a statement like this if you're not citing every study.

The sentence is written more accurately now: "a large body of research deals with global-scale climate signals and beach change (e.g., Masselink et al., 2023). However, only a small number of studies have associated synoptic-scale CPs with wave climates along the shores of oceans or lakes…".

L79: inferred

Corrected.

L172: what are ARSTs?

ARSTs = Active Red Sea Troughs. Now it is defined in the right place.

L196: these are very steep waves.

That is true. The word 'steep' was added to the description of the properties of the waves along with the sentence: 'the high viscosity\density of the brine (Weisbrod et al., 2016) may explain the steepness of the observed wave.'

References:

Enzel, Y., Mushkin, A., Groisman, M., Calvo, R., Eyal, H., and Lensky, N.: The modern wave-induced coastal staircase morphology along the western shores of the Dead Sea, Geomorphology, 408, 108237, https://doi.org/10.1016/j.geomorph.2022.108237, 2022.

Masselink, G., Scott, T., Poate, T., Stokes, C., Wiggins, M., Valiente, N., and Konstantinou, A.: Tale of two beaches: correlation between decadal beach dynamics and climate indices, in: Coastal Sediments 2023: The Proceedings of the Coastal Sediments 2023, World Scientific, 337–350, https://doi.org/https://doi.org/10.1142/9789811275135_0031, 2023.

Nienhuis, J. H., Ashton, A. D., and Giosan, L.: Littoral steering of deltaic channels, Earth Planet. Sci. Lett., 453, 204–214, 2016.

Weisbrod, N., Yechieli, Y., Shandalov, S., and Lensky, N.: On the viscosity of natural hyper-saline solutions and its importance: The Dead Sea brines, J. Hydrol., 532, 46–51, https://doi.org/https://doi.org/10.1016/j.jhydrol.2015.11.036, 2016.